# Dosage differences in *12-OXOPHYTODIENOATE REDUCTASE* genes modulate wheat root growth

Gilad Gabay [1], Hanchao Wang[1,2], Junli Zhang [1], Jorge I. Moriconi [3,4], German F. Burguener [1], Leonardo D. Gualano [3,4], Tyson Howell [1], Adam Lukaszewski [5], Brian Staskawicz[6], Myeong-Je Cho [6], Jaclyn Tanaka [6], Tzion Fahima[2], Haiyan Ke[7], Katayoon Dehesh[7], Guo-Liang Zhang[8], Jin-Ying Gou [8,9], Mats Hamberg [10], Guillermo E. Santa-María [3,4] ✉ & Jorge Dubcovsky [1,11] ✉

Wheat, an essential crop for global food security, is well adapted to a wide variety of soils. However, the gene networks shaping different root architectures remain poorly understood. We report here that dosage differences in a cluster of monocot-specific *12-OXOPHYTODIENOATE REDUCTASE* genes from subfamily III (*OPRIII*) modulate key differences in wheat root architecture, which are associated with grain yield under water-limited conditions. Wheat plants with loss-of-function mutations in *OPRIII* show longer seminal roots, whereas increased *OPRIII* dosage or transgenic over-expression result in reduced seminal root growth, precocious development of lateral roots and increased jasmonic acid (JA and JA-Ile). Pharmacological inhibition of JA-biosynthesis abolishes root length differences, consistent with a JA-mediated mechanism. Transcriptome analyses of transgenic and wild-type lines show significant enriched JA-biosynthetic and reactive oxygen species (ROS) pathways, which parallel changes in ROS distribution. *OPRIII* genes provide a useful entry point to engineer root architecture in wheat and other cereals.

Further increases in wheat grain yield are required to feed a growing population, but losses generated by water stress are increasing with global warming, and are eroding progress in other areas of wheat improvement[1]. Root depth and biomass distribution in the soil profile are critical traits for adaptation to water stress and have been prioritized for improving drought resilience in wheat[2–5]. However, the gene networks that regulate these traits in wheat remain largely unknown. This has prompted new efforts to understand and modify wheat root architecture to optimize water acquisition in both common (*Triticum aestivum*, genomes AABBDD) and durum wheat (*T. turgidum* ssp. *durum*, genomes AABB)[4].

[1]Department of Plant Sciences, University of California, Davis, CA 95616, USA. [2]Department of Evolutionary and Environmental Biology, Institute of Evolution, University of Haifa, Haifa 3498838, Israel. [3]Instituto Tecnológico de Chascomús (INTECH), Consejo Nacional de Investigaciones Científicas y Técnicas (CONICET), B7130 Chascomús, Buenos Aires, Argentina. [4]Escuela de Bio y Nanotecnologías (EByN), Universidad Nacional de San Martín (UNSAM), B1650 San Martín, Buenos Aires, Argentina. [5]Department of Botany & Plant Sciences, University of California, Riverside, CA 92521, USA. [6]Innovative Genomics Institute, University of California, Berkeley, CA 94704, USA. [7]Institute for Integrative Genome Biology, University of California, Riverside, CA 92521, USA. [8]Institute of Plant Biology, School of Life Sciences, Fudan University, Shanghai 200438, China. [9]Key Laboratory of Crop Heterosis and Utilization (MOE) and Beijing Key Laboratory of Crop Genetic Improvement, China Agricultural University, Beijing 100193, China. [10]Division of Physiological Chemistry II, Department of Medical Biochemistry and Biophysics, Karolinska Institutet, S-171 77 Stockholm, Sweden. [11]Howard Hughes Medical Institute, Chevy Chase, MD 20815, USA. ✉e-mail: gsantama@intech.gov.ar; jdubcovsky@ucdavis.edu

A tested source for improving these traits in wheat is the intro-gression of the short arm of rye (*Secale cereale* L.) chromosome one (1RS) into common wheat (henceforth 1RS.1BL), which induces higher root biomass and confers a yield advantage under drought stress[6–9]. Unfortunately, this translocation reduces bread baking quality[10,11]. To eliminate the negative effect on quality, a recombinant 1RS chromo-some with two wheat 1BS interstitial introgressions (henceforth 1WW) was developed twenty years ago in the wheat cultivar Pavon[12]. We introgressed the engineered 1WW chromosome into the 1RS.1BL cv.

Hahn by six backcrosses, and then generated three lines with either the complete 1RS (1RS), the proximal 1BS introgression removing the rye secalins (1WR), or the distal 1BS introgression restoring the *Gli-B1 / Glu-B3* locus (1RW, Fig. 1A). Using the wheat Illumina 90 K SNP iSelect array, we demonstrated that the 1RS and 1RW lines were 99.3% identical[13].

In large physiological and agronomic field experiments replicated over four years (2008–2012), in two locations, and under both full-irrigation and terminal drought, lines carrying the complete 1RS arm showed significantly higher grain yield relative to 1RW (up to 40%

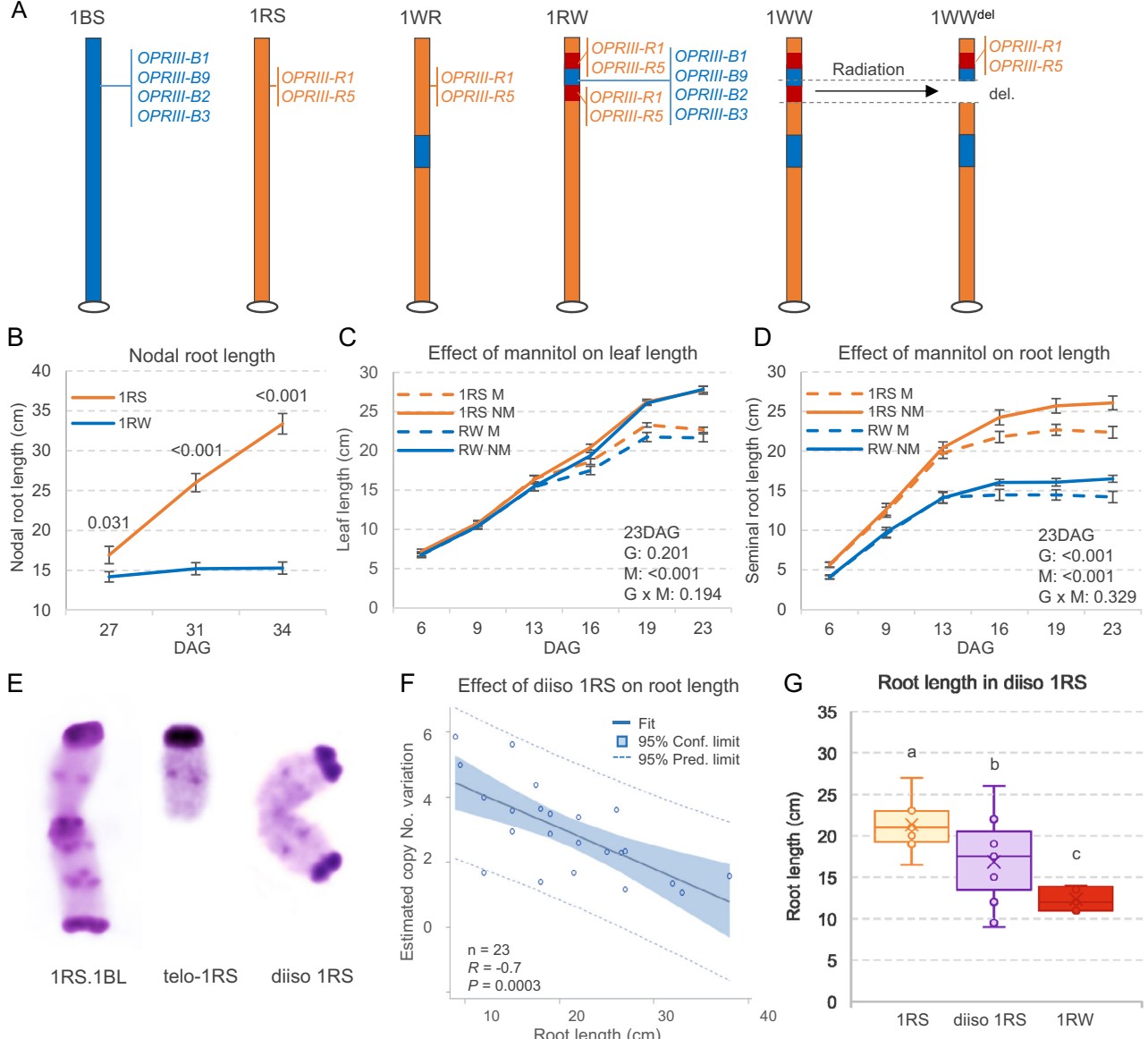

**Fig. 1 | Wheat-rye genetic stocks and their effects on seminal root length.**
**A** Genetic stocks used in this project. The hexaploid wheat variety Hahn has a complete rye 1RS arm replacing the wheat 1BS arm. Line 1WR has a proximal 1BS segment introgressed in 1RS, whereas 1RW has a distal 1BS introgression. Line 1WW has both wheat introgressions and 1WW^del has a 4.9 Mb deletion including part of 1BS and 1RS. Wheat chromatin is indicated in blue, rye in orange, and the duplicated region in rye in red. The effect of these structural changes on *OPRIII* gene dosage is presented as an example. **B** Differences in nodal root growth between 1RS and 1RW (*n* = 10). Exact *P* values based on two-tailed *t* tests at each time point. Effect of mannitol on (**C**) leaf growth and (**D**) seminal root growth in 1RS and 1RW (*n* = 15–18). Exact *P* values in genotype x mannitol factorial ANOVA at 23 DAG are indicated for G = genotype, M = mannitol, NM = no mannitol, and G x M = interaction. **B–D** Data

are presented as means values ± SEM. **E** C-banding of the 1RS.1BL translocation, the 1RS telocentric and a diiso 1RS chromosome. Hahn-1RS plants homozygous for diiso 1RS have four extra 1RS arms and, therefore, triple dosage of genes.
**F** Regression between 1RS copy number variation (CNV) and root length in hydroponic tanks 19 DAG (*n* = 23). The central line indicates the least squares best fit line. **G** Seminal root length of 13 diiso 1RS lines with at least one extra 1RS arm (total 3–6 1RS arms, average 4.0 ± 0.3) relative to 1RS plants (*n* = 8, two 1RS arms) and 1RW plants with a triplicated 1RS/1BS candidate gene region (*n* = 8, 4 1RS + 2 1BS copies). Different letters indicate significant differences based on two-tailed Tukey test (*P* < 0.05). The boxes show the range from first to third quartiles divided by the median. The whiskers span from the minimum to the maximum observation and circles indicate individual data. Source data are provided as a Source Data file.

**Table 1 | Comparison of root and aerial parts of 1RS and 1RW at 26 DAG**

|  | Seminal root length (cm) | Root dry weight (g) | Number of leaves | Leaf[a] length (mm) | Shoot dry weight (g) |
|---|---|---|---|---|---|
| 1RS[b] | 43.21 ± 0.94 | 0.043 ± 0.002 | 4.42 ± 0.03 | 313.7 ± 5.1 | 0.138 ± 0.007 |
| 1RW[b] | 16.94 ± 0.79 | 0.039 ± 0.003 | 4.40 ± 0.06 | 312.8 ± 5.4 | 0.128 ± 0.008 |
| Genotype P | <0.0001 | 0.1788 | 0.6542 | 0.8947 | 0.2626 |
| Experiment P | 0.1725 | <0.0001 | <0.0001 | 0.0062 | <0.0001 |
| Normality of residuals P | 0.1387 | 0.384 | 0.1104 | 0.844 | 0.6895 |

Combined ANOVA for two experiments using experiments as blocks.
Source data are provided as a Source Data file.
[a]Measured in youngest fully expanded leaf.
[b]$n = 10$ per genotype /per experiment; total $N = 40$.

higher under terminal water stress)[13]. The 1RS lines showed higher carbon isotope discrimination and increased stomatal conductance, indicating improved access to residual water in the soil. Increased water content was also evident in replicated split-plot field experiments where 1RS plants showed improved water indexes and absence of rolled leaves and dry leaf tips that were evident some years in the adjacent 1RW plots[13].

In a subsequent field study, we excavated three ~2 m deep trenches cutting perpendicular across the middle of the plots in three blocks including the different genotypes and took horizontal soil core samples from the center of each plot at 20 cm intervals[14]. These data confirmed that the 1RS lines have a higher root density, with roots that reach deeper in the soil and can access more water than the 1RW lines[14]. The differences in wheat architecture were also evident under hydroponic conditions, where the elongation of the 1RW seminal roots decreased from nine days after germination (DAG) and almost stopped by ~16 DAG while the 1RS roots continue to grow. The 1RW roots also showed lateral roots forming close to the seminal root apical meristem (RAM), a phenotype that was not observed in the near isogenic sister lines with the complete 1RS arm[14]. In summary, these results indicated that the small wheat introgression in 1RW was associated with drastic changes in root architecture both in short term hydroponic assays and in field experiments across the growing season.

More recent exome sequencing of the different lines showed that the 1RW chromosome has a complex structure generated by structural differences between the 1RS and 1BS distal chromosome regions[15–17]. The 1RW chromosome has a duplicated 1RS region (7.0 Mb) flanking an insertion of a colinear region from wheat chromosome arm 1BS (4.8 Mb), which resulted in increased dosage of the genes included in the triplicated region[17]. Figure 1A shows the effect of these structural changes on a gene family located within the duplicated region.

The previous study[17] also showed that different 1RS-1BS recombinant lines with longer distal regions from 1BS or 1RS (but without the duplication present in 1RW) have long seminal roots, suggesting that changes in gene dosage are responsible for the shorter seminal roots in 1RW rather than specific 1BS or 1RS genes. This hypothesis was also supported by the intermediate seminal-root length of heterozygous 1RW plants[17], and by the restoration of normal root growth in a 1WW line carrying a small deletion that eliminated part of the distal 1BS introgression and the proximal duplicated 1RS segment[17] (Fig. 1A). This radiation mutant further delimited a candidate gene region including 38 wheat and rye duplicated orthologs, but the causal gene was not identified.

In this study, we present the functional validation of the genes responsible for the differences in root development using CRISPR-Cas9 induced mutants and transgenic lines. We also demonstrate that these genes encode cytoplasmic functional enzymes involved in the biosynthesis of JA-Ile, and that a JA biosynthetic inhibitor can abolish the differences between 1RS and 1RW in root architecture. Finally, we present high-throughput transcriptomic data for the isogenic and transgenic lines that reveal the pathways affected by the differences in dosage or expression levels of these genes.

## Results

### Effect of 1BS/1RS duplication on nodal roots and leaves

To prioritize the candidate genes for functional validation, we performed additional characterizations of the effects of the 1BS introgression and 1RS duplication on nodal roots growth and aerial tissues dry weight. Nodal roots started to appear at 18 DAG and initially grew similarly in 1RS and 1RW. However, by 27 DAG the 1RS longest nodal roots were 2.7 cm longer ($P = 0.031$) than the equivalent root in 1RW, a difference that increased to 10.8 cm at 31 DAG and to 18.1 cm at 34 DAG ($P < 0.001$, Fig. 1B). These results indicate that the 1RW nodal roots experience a similar developmentally-regulated growth arrest as the seminal roots. Taken together, these results are consistent with the higher root density and increased water access of the 1RS plants relative to 1RW in the field experiments[14].

In replicated hydroponic experiments showing highly significant differences in seminal root length ($P < 0.0001$) between 1RS and 1RW, we analyzed the aerial part of the plants. A combined ANOVA for two experiments showed no significant differences in leaf number, length of the youngest fully expanded leaf or aerial dry weight at 26 DAG (Table 1). We observed a slight decrease in both aerial and root dry weight in 1RW relative to 1RS (average 7.2% and 8.0%, respectively), which was marginally significant in one of the experiments ($P = 0.035$ and $P = 0.036$, respectively) but not in the combined analyses (Table 1). Based on these results, we concluded that the differences between the 1RS and 1RW genotypes mainly involve changes in root elongation and therefore, we prioritized candidate genes expressed predominantly in roots for functional characterization.

To test if 1RS and 1RW exhibit differential physiological responses under osmotic stress, we compared them in hydroponic experiments with and without mannitol (200 mM). We found no significant differences between 1RS and 1RW for leaf length (Fig. 1C) and relative water content (source data of Fig. 1), but detected significant reductions in root length (Fig. 1D) and root and shoot dry weight (source data of Fig. 1) in 1RW relative to 1RS. All traits showed highly significant differences for the mannitol treatment, but no significant genotype x mannitol interactions (source data of Fig. 1), indicating that both genotypes respond similarly to the osmotic stress induced by mannitol (Fig. 1D).

Finally, we tested the effect of the 1RS duplication on seminal root architecture in the absence of the 1BS segment introgression. To do this, we developed a diisosomic 1RS addition line with two fused 1RS arms into Hahn-1RS (diiso 1RS, Fig. 1E). The seminal roots from plants carrying extra 1RS arms were significantly shorter than those from the wildtype (Fig. 1F, G and the associated source data), confirming that duplicated 1RS gene dosage is sufficient to induce shorter seminal roots.

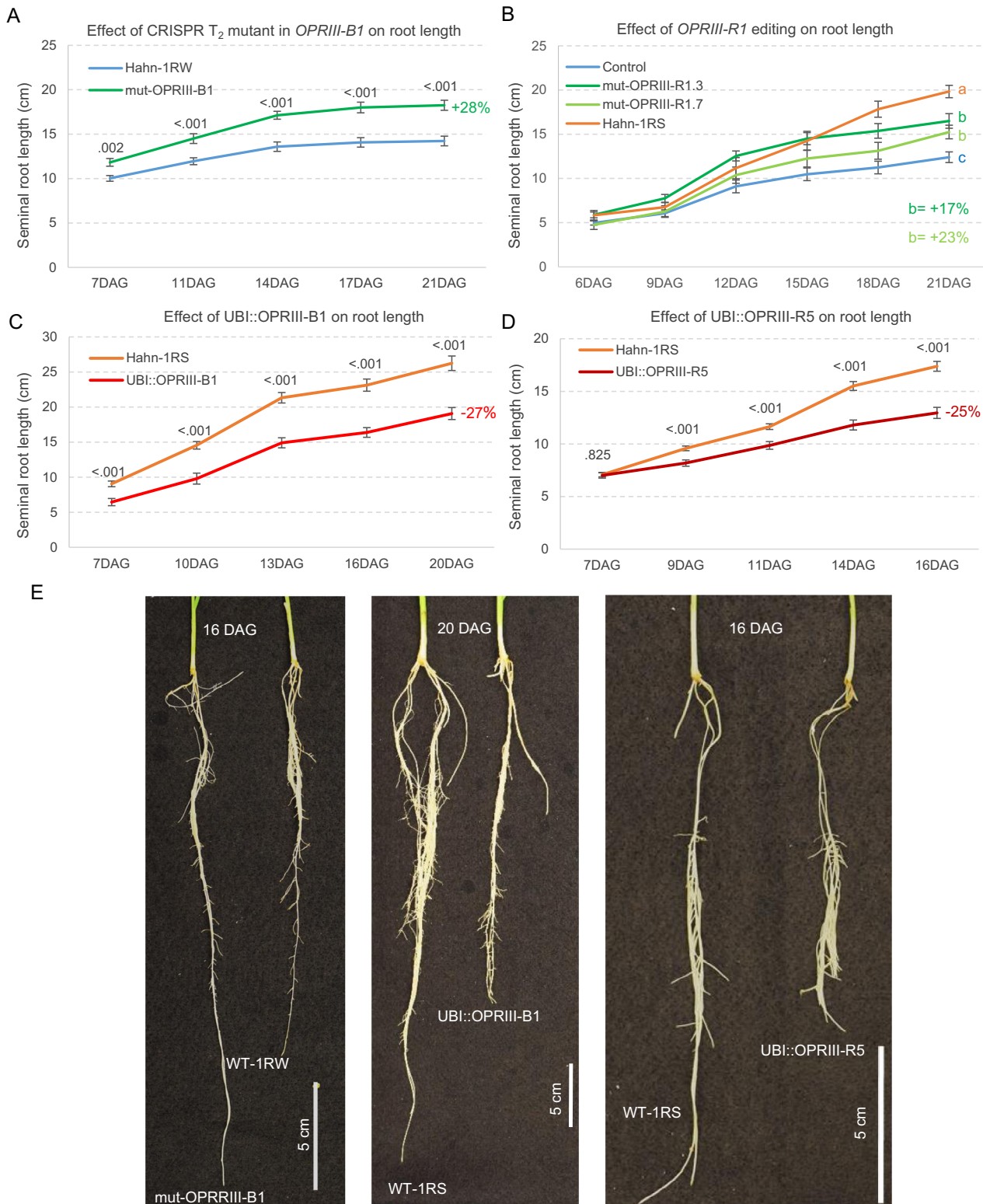

**Fig. 2 | Effect of loss-of-function mutations and over-expression of *OPRIII* genes on root length.** **A** Effect of 1RW T$_2$ sister lines with and without a 32-bp deletion in *OPRIII-B1* on seminal root length (*n* = 20–22). Exact *P* values based on two-tailed *t* tests are indicated at each time point. **B** Effect of 1RW T$_2$ lines with at least one copy of a 3-bp (*n* = 12) or a 7-bp (*n* = 8) deletions in *OPRIII-R1* (*R1.3* and *R1.7*) on root seminal length. Control = combined 1RW parental line and 1RW transgenic sister lines without the deletion (*n* = 25), 1RS = Hahn-1RS with no *OPRIII* duplication

(*n* = 6). Different letters indicate significant differences in two-tailed Tukey test at 21 DAG (*P* < 0.05). Transgenic Hahn-1RS lines constitutively expressing (**C**) UBI::OPRIII-B1 (*n* = 20) or (**D**) UBI::OPRIII-R5 (*n* = 48). Exact *P* values based on two-tailed *t* tests at each time point. **A**–**D** Data are presented as means values ± SEM. **E** Images of roots lengths in the same genotype comparisons as in (**A**), (**C**) and (**D**). Source data are provided as a Source Data file.

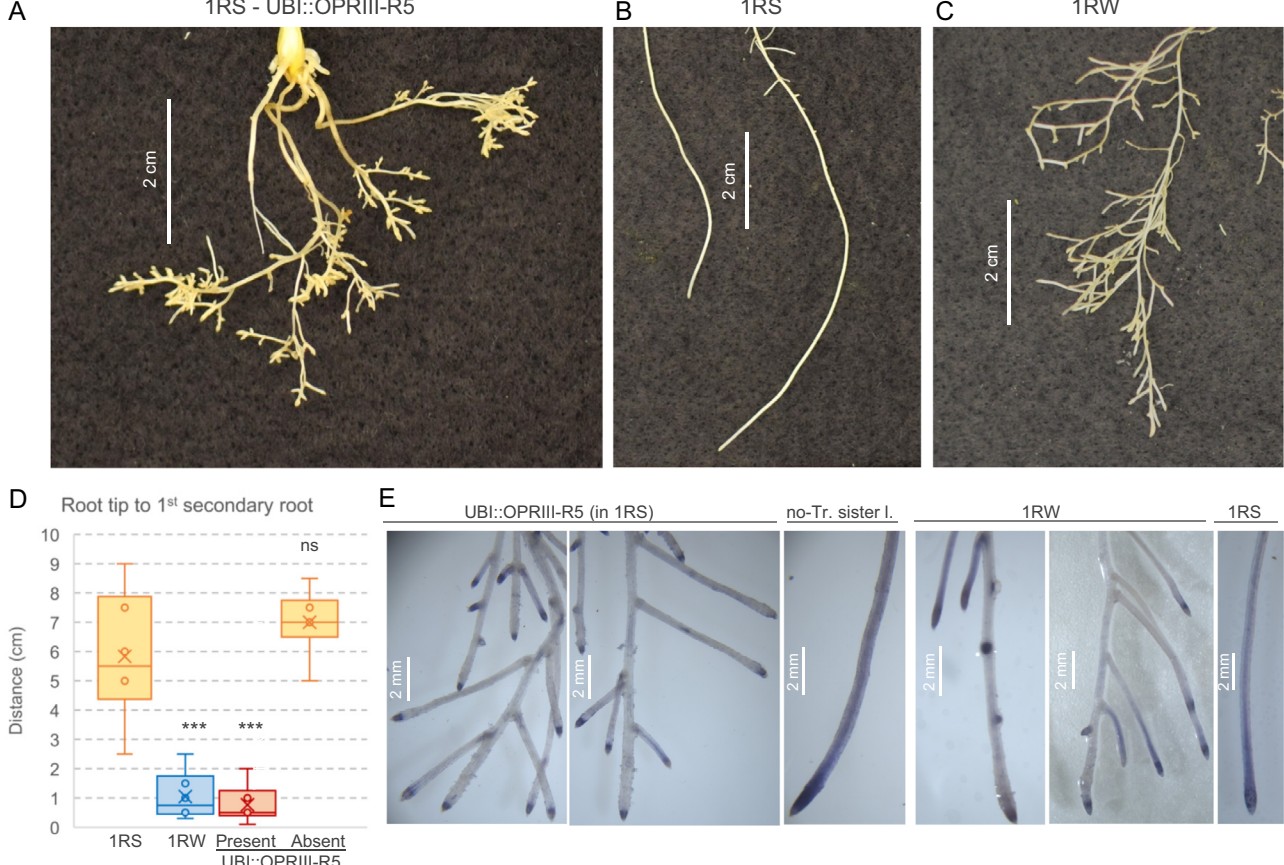

**Fig. 3 | Effect of over-expression of *OPRIII* genes on root architectures. A–C** Last 5 cm of the seminal roots showing different branching patterns. **A** UBI::OPRIII-R5, (**B**) 1RS and (**C**) 1RW. **D** Distance between the root tip and the first secondary root in 1RS, 1RS$^{RW}$ and UBI::OPRIII-R5 sister lines with and without the transgene (*n* = 6). The boxes show the range from first to third quartiles divided by the median. The whiskers span from the minimum to the maximum observation and circles indicate individual data. *P* values based on two-sided Dunnett tests relative to 1RS control: ns = not significant, *** = *P* < 0.001. **E** Seminal roots from UBI::OPRIII-R5 sister lines, 1RW and 1RS stained with NBT to visualize the distribution of ROS. UBI::OPRIII-R5 = transgenic line over-expressing *OPRIII-R5* in a 1RS background, no-Tr. Sister l.= sister line without the transgene, 1RW = Hahn line with a triple dose of the *OPRIII* genes, and 1RS = Hahn line with a single dose of the *OPRIII* genes. These experiments were replicated twice with identical results. Source Data are provided as a Source Data file.

## Characterization of the 12-OXOPHYTODIENOATE REDUCTASE SUBFAMILY III (OPRIII) candidate genes

Among the 14 high-confidence genes annotated in the deleted 1BS-1RS region and expressed in the roots[17], we prioritized four *OPRIII* genes for functional validation. These genes encode enzymes involved in the biosynthesis of JA[18], whose JA-Ile conjugate is an active phytohormone proposed to modify root architecture in Arabidopsis[19] and rice[20], as well as responses to drought[21,22]. The *OPR* gene family expanded in the vascular plants, where five well-conserved subfamilies (I-V) have been reported[23,24]. The *OPR* genes described here belong to the monocot-specific subfamily III and have been designated as *12-OXOPHYTODIENOATE REDUCTASE SUBFAMILY III (OPRIII)* in a previous phylogenetic study of the wheat *OPR* genes[24,25]. Gene names, coordinates and predicted proteins used in this study are described in source data of Supplementary Figs. 1 and 2. The wheat *OPRIII* genes are clustered mainly in the short arms of homoeologous group 1 (*OPRIII1* to *OPRIII9*) and 7 (*OPRIII12* and *OPRIII13*). The latter cluster is colinear with a rice region on chromosome 6 that includes six tightly linked *OPR* genes from the same subfamily (*OPR6.1-OPR6.6*)[18,24], which are more similar to the wheat *OPRIII* genes on chromosome 7 than to those located on chromosome 1 based on the phylogenetic tree in Supplementary Figs. 1 and 2. The rice region colinear with the wheat *OPRIII* cluster on chromosome arm 1BS shows no *OPR* genes, suggesting that the *OPRIII* expansion on the short arm of wheat homoeologous group

1 occurred in the temperate grasses or the Triticeae lineage. Based on the same phylogenetic tree, the rye genes on chromosome arm 1RS were designated as *OPRIII-R2* and *OPRIII-R5*. In addition to the two major clusters in groups 1 and 7, wheat has a few additional *OPRIII* genes dispersed in other four chromosomes[24].

A transcriptome analysis of the seminal root terminal region (1 cm) in 1RS and 1RW at 6 and 16 DAG (source data of Fig. 6) revealed that most *OPRIII* genes are expressed at significantly higher levels at 6 DAG than at 16 DAG, a result confirmed by qRT-PCR (Supplementary Fig. 3). As expected, the duplicated rye *OPRIII* genes showed higher expression in 1RW than in 1RS (Supplementary Fig. 3), and *OPRIII-B1* (present in the 1BS introgression, Fig. 1E) was detected in 1RW but not in 1RS (Supplementary Fig. 3 and the associated source data). These combined changes contributed to an overall higher *OPRIII* expression in 1RW than in 1RS.

## Functional validation of the *OPRIII* candidate genes

To validate the role of the *OPRIII* genes on seminal root length, we targeted the duplicated wheat and rye *OPRIII* genes in 1RW for CRISPR-Cas9 editing. We found a 32-bp frame-shift induced deletion in *OPRIII-B1* exon 2 (mut-OPRIII-B1) and two deletions in *OPRIII-R1* of 7-bp (mut-OPRIII-R1.7) and 3-bp (mut-OPRIII-R1.3). All three mutations result in premature stop codons and truncated proteins that lack 75–98% of the amino acids, so they are almost certainly not functional. The T$_2$ plants

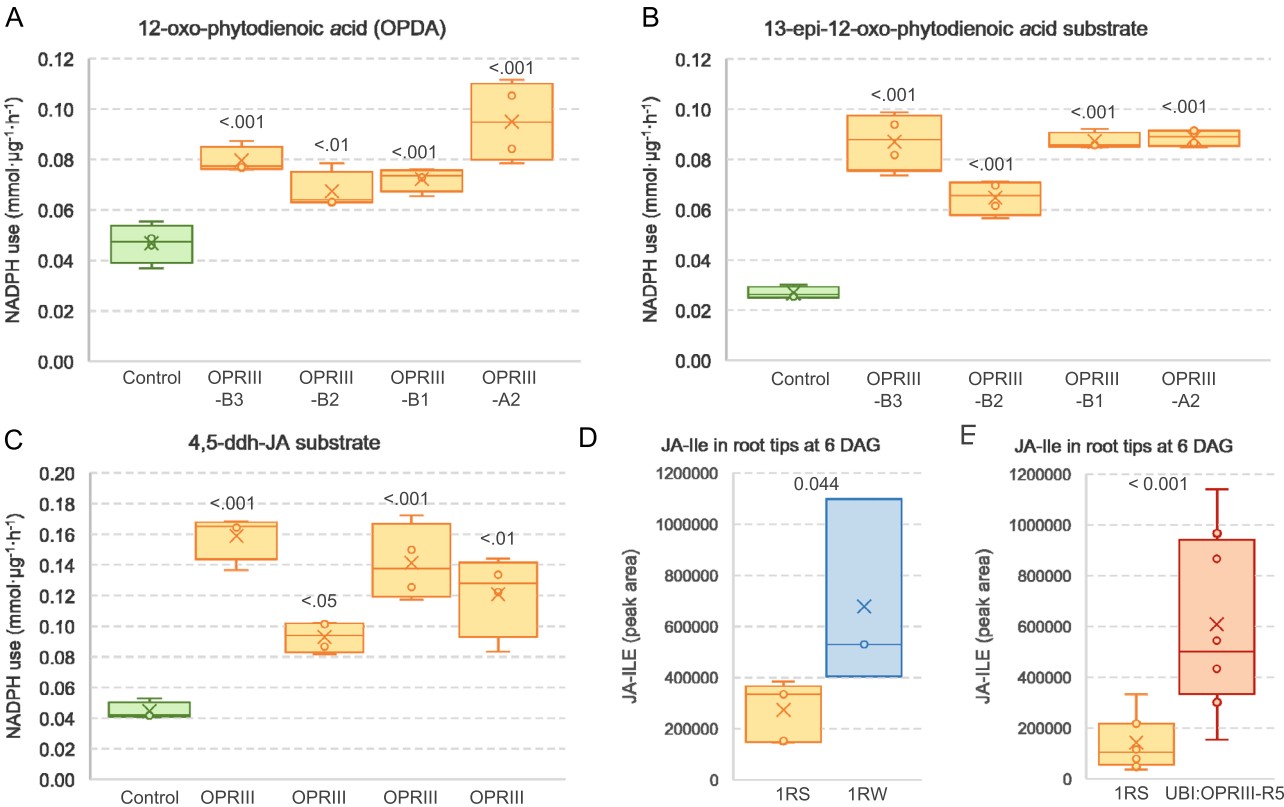

**Fig. 4 | OPRIII role in the biosynthesis of JA. A–C** Consumption of NADPH (nicotinamide adenine dinucleotide phosphate) by different OPRIII proteins using different substrates. **A** 12-oxo-phytodienoic acid (OPDA), (**B**) 13-epi-2-oxo-phytodienoic acid (**C**) 4,5-ddh-JA. A boiled mixture of recombinant proteins was used as the negative control. **A–C** *P* values based on Dunnett tests relative to control ($n = 4$). **D** JA-Ile content in root tips of 1RS ($n = 5$) and 1RW ($n = 3$) at 6 DAG. **E** JA-Ile content

in root tips of 1RS ($n = 8$) and UBI::OPRIII-R5 ($n = 8$) at 6 DAG. *P* values based on ANOVA. The boxes show the range from first to third quartiles divided by the median. The whiskers span from the minimum to the maximum observation and circles indicate individual data. All statistical tests are two-sided. **A–E** Source Data are provided as a Source Data file.

homozygous for the mut-OPRIII-B1 deletion (without the transgene) showed 28% longer seminal roots than the sister 1RW plants ($P < 0.0001$, Fig. 2A, source data of Fig. 2). A replicated experiment using $T_3$ plants showed similar results (Supplementary Fig. 5 and the associated source data). The $T_2$ plants carrying at least one copy of the mut-OPRIII-R1.7 or mut-OPRIII-R1.3 deletions also showed significantly longer roots than the 1RW sister lines (Fig. 2B, 17–23% increase, source data of Fig. 2). Taken together, these results demonstrate that reduced *OPRIII* dosage is associated with increased seminal root length.

We then developed transgenic Hahn-1RS and Fielder lines constitutively expressing either wheat *OPRIII-B1* (UBI::OPRIII-B1) or rye *OPRIII-R5* (UBI::OPRIII-R5) genes driven by the maize *UBIQUITIN* promoter (Supplementary Fig. 4). We confirmed that both transgenes were expressed in leaves, a tissue where the endogenous *OPRIII* genes were not detected in the non-transgenic controls (source data of Fig. 2). At the end of the hydroponic experiments, the seminal roots of the transgenic plants were 18–28% shorter than those of their non-transgenic sister lines in both genetic backgrounds (Fig. 2C, D and Supplementary Fig. 6A, B). These differences varied in time and resulted in highly significant genotype x time interactions in repeated measure ANOVAs (source data of Fig. 2 and Supplementary Fig. 6), suggesting that the effects of *OPRIII* constitutive expression on seminal root elongation vary with root development. Since ectopic expression of these two *OPRIII* genes resulted in significant reductions in leaf length (9-10%) and aerial dry weight (30–36%) at 23 DAG (source data of Fig. 2), we cannot rule out an indirect effect of these changes on the observed differences in the roots of the transgenic plants. However,

when driven by their natural promoters, the *OPRIII* genes are expressed mainly in the roots, suggesting a primary effect in this tissue.

In both UBI::OPRIII-R5 and 1RW, once seminal root growth was arrested, secondary roots emerged significantly closer to the RAM than in 1RS (Fig. 3A–D and the associated source data). Staining of these roots with nitro blue tetrazolium (NBT) suggested that ROS were more restricted to the distal root region in 1RW and UBI::OPRIII-R5 than in 1RS (Fig. 3E), as described for 1RW[14]. The significant effects of the loss-of-function mutation and over-expression of *OPRIII* genes on seminal root length and architecture demonstrate that the *OPRIII* genes are responsible for the different root phenotypes in 1RS and 1RW isogenic lines, and likely for their different performance in the field under water stress[13].

### Enzymatic activity and subcellular localization of the OPRIII proteins

We then confirmed that wheat *OPRIII-B1*, *OPRIII-B2*, *OPRIII-B3*, and *OPRIII-A2* encode functional 12-oxophytodienoate reductase enzymes. Expressed proteins from all four genes were able to significantly reduce 12-oxo-phytodienoic acid (OPDA), 13-epi-12-OPDA and 4,5-ddh-JA substrates using NADPH (Fig. 4A–C and the associated source data), as shown in published studies for other OPR enzymes[18,26].

In Arabidopsis, OPR3 (Group II) processes OPDA into OPC8 in the peroxisome, whereas OPR2 (Group I) processes 4,5-ddh-JA into (-)-JA in the cytoplasm[27]. Since wheat OPRIII enzymes were able to use both OPDA and 4,5-ddh-JA as substrates in vitro, we performed a subcellular localization experiment in tobacco leaves using OPRIII-GFP fusions and

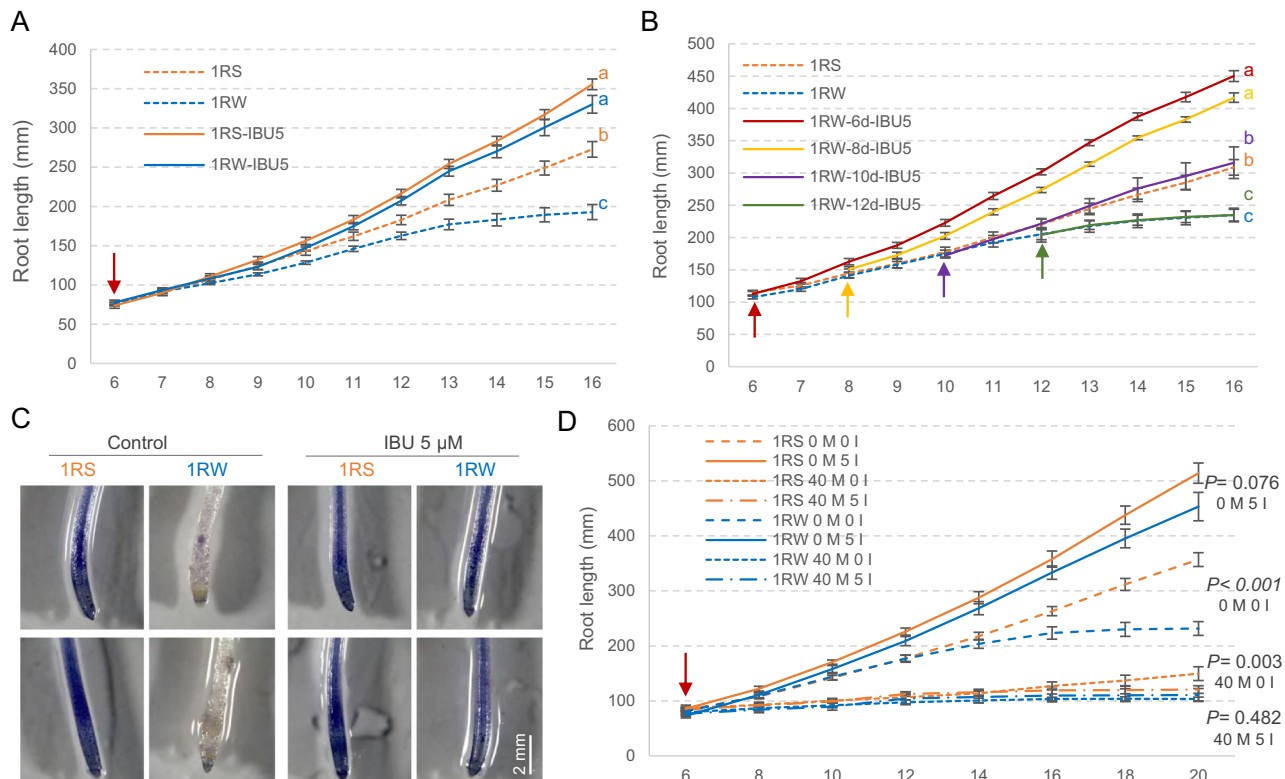

**Fig. 5 | Effect of Ibuprofen (IBU) on root growth. A** Average root length of 1RS and 1RW ($n = 9–10$) with and without 5 µM-IBU added at day 6 (red arrow). **B** Effect of 5 µM-IBU added at 6, 8, 10, and 12 DAG (arrows) on 1RW root length ($n = 8$). **A**, **B** Different letters indicate significant differences at 16 DAG based on two-sided Tukey tests ($P < 0.05$). **C** Effect of 5 µM-IBU on ROS distribution along roots (as determined with NBT). **D** Effect of combined 5 µM-IBU and 40 µM-MeJA on root length ($n = 8$). M = MeJA, I = IBU, Numbers (0, 5, 40) = µM concentrations. Exact $P$ values based on two-tailed $t$ tests between 1RS and 1RW under identical IBU and MeJA concentrations. Data are presented as means values ± SEM. Source data are provided as a Source Data file.

the mCherry peroxisome marker CD3-983[28]. Our results indicate that OPRIII-B1, OPRIII-B2, OPRIII-B3 and OPRIII-A2 are all located in the nucleus and cytoplasm outside the peroxisome, similarly to what was reported for maize OPR2 (Group III) in Tolley et al.[29] (Supplementary Fig. 7).

### Effect of the *OPRIII* genes on JA-Ile concentrations and their role in root differences

We measured levels of the bioactive JA-Ile and its precursors JA and OPDA in the terminal 1 cm of the seminal roots at 6 DAG. We observed a 2.5 to 3.3-fold increase in JA and JA-Ile concentrations in 1RW relative to 1RS ($P < 0.05$) and a 4.3 to 4.8-fold increase in UBI::OPRIII-R5 relative to 1RS ($P < 0.01$, Fig. 4D, E and the associated source data). These results demonstrate that increases in *OPRIII* gene dosage or expression are associated with increases in JA and JA-Ile in early seminal root development. We did not find significant differences in OPDA among the three genotypes (source data of Fig. 4).

To explore the effect of the increases in JA-Ile on wheat genes known to be involved in the JA-signaling pathway (*COI1, JAZ1, JAM3* and *MYC2*, Supplementary Fig. 8)[30], we compared the root transcriptome data from 1RW and UBI::OPRIII-R5 with 1RS (source data of Fig. 6). For each gene, we analyzed simultaneously the three homoeologs in a factorial ANOVA including genotypes and homoeologs as factors and RNA-seq samples as replications. No significant interactions between homeologs and genotypes were detected, indicating similar responses among homoeologs. All four genes showed significant differences in expression between 1RS and at least one of the two genotypes with increased JA-Ile (Supplementary Fig. 8). We also observed a significant increase in the transcripts of the JA-regulated genes *PLETHORA 1*

(*PLT1*), *PLT3* and *PLT5* in 1RW and UBI::OPRIII-R5 relative to 1RS at *16 DAG* but not at 6 DAG (Supplementary Fig. 8). These results support a role of the *OPRIII* genes on the JA-Ile signaling pathway, which is known to be involved in the regulation of root architecture[30].

To pharmacologically test if the differences in JA-Ile were responsible for the reduced seminal root growth of the 1RW lines, we used Ibuprofen (IBU), a known inhibitor of the JA biosynthetic pathway[31]. The addition of 5 µM-IBU in the hydroponic culture from 6 DAG resulted in similar seminal root elongation in 1RS and 1RW by the end of the experiment at 16 DAG (Fig. 5A and the associated source data). The addition of 5 µM-IBU also accelerated seminal root elongation in 1RW when added from 8 or 10 DAG, but not from 12 DAG (Fig. 5B and the associated source data), suggesting that IBU is no longer effective when the JA-Ile signaling cascade is already induced. The addition of 5 µM-IBU also eliminated the differences in the distribution of ROS between 1RW and 1RS (Fig. 5C). These results indicate that differences in both root length and ROS distribution between 1RS and 1RW are likely mediated by changes in JA-Ile.

We also analyzed the effect of combined levels of 5 µM-IBU and 40 µM-MeJA applied from 6 DAG on 1RS and 1RW root length. Plants treated only with IBU showed similar results to those in the previous experiment, whereas those treated only with MeJA showed inhibited root growth (Fig. 5D). The MeJA treatment alone reduced but not eliminated the differences between 1RS and 1RW genotypes. At 20 DAG, the differences between genotypes were significant ($P < 0.01$, source data of Fig. 5) in the MeJA treatment but not significant in the combined MeJA-IBU treatment (Fig. 5D). A three-way ANOVA showed significant differences for the three

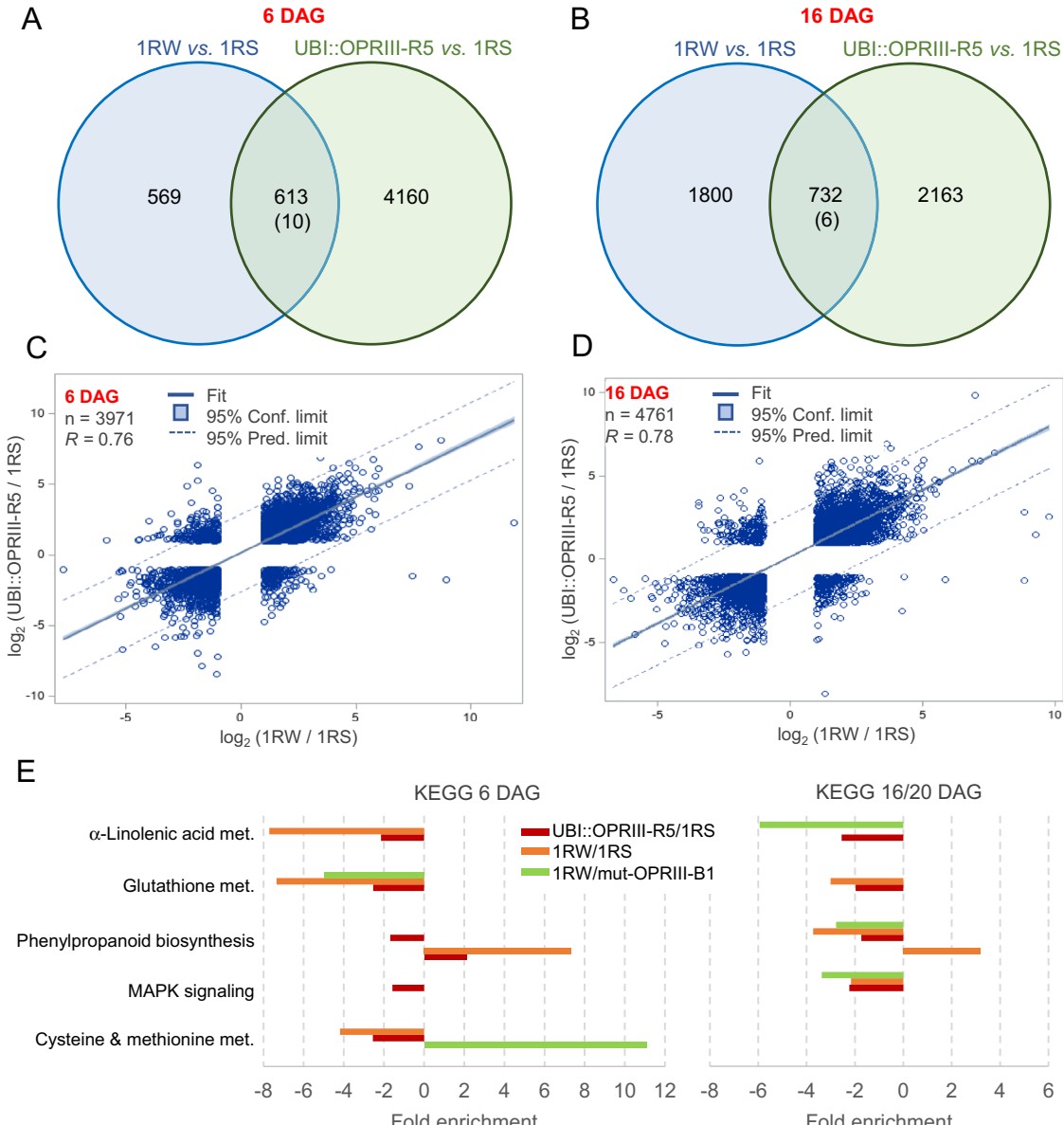

**Fig. 6 | Transcriptome profiles of root tips in different genotypes. A–D** Differentially expressed genes (DEGs, FDR < 0.05) in comparisons between 1RW *vs*. 1RS and UBI::OPRIII-R5 *vs*. 1RS at 6 and 16 DAG. **A, B** Venn Diagrams. The number in the intersection indicates common DEGs in the same direction, and the number in parenthesis in opposite directions. **C, D** Regression between log$_2$ expression ratios 1RW/1RS and UBI::OPRIII-R5/1RS for genes with more than two-fold changes in expression relative to 1RS (log$_2$ fold < −1 or > +1) and at least one replication >0. Regressions and $R^2$ were calculated based on 3971 genes in (**C**) and 4,761 genes in (**D**). The central line indicates the least squares best fit line. The significant

regressions ($P < 0.001$) indicate similar changes in expression in 1RW and UBI::O-PRIII-R5 relative to 1RS. **E** Significant enriched pathways in KEGG analyses in three transcriptome comparisons: 1RW / 1RS (6 and 16DAG), UBI::OPRIII-R5 / 1RS (6 and 16DAG), and 1RW / mut-OPRIII-B1 (32-bp deletion, 6 and 20 DAG). Only pathways differentially enriched ($P < 0.05$, based in 2 × 2 contingency tables) in all three comparisons were included (in any combination across days). Separate analyses were performed for the early and late time points and for the upregulated (ratio >1.0) and downregulated (ratio <1.0). Source data are provided as a Source Data file.

main factors (Genotype, MeJA and IBU) and the three two-way interactions (source data of Fig. 5). The significant genotype x MeJA ($P < 0.001$) and genotype x IBU ($P < 0.05$) interactions are consistent with the hypothesis that the root length differences between 1RS and 1RW are associated with changes in the JA biosynthetic / signaling pathway. These results are also consistent with the observed differences in endogenous JA/JA-Ile levels in the roots between genotypes and the demonstrated role of the *OPRIII* genes on the differences in root length in the transgenic plants.

### Effects of *OPRIII* increased dosage or expression on the transcriptome

A comparison of the transcriptomes of the distal 1 cm of the seminal roots between 1RS and both 1RW and UBI::OPRIII-R5 (source data of Fig. 6, Supplementary Fig. 9) revealed a large number of differentially expressed genes (DEGs) at both 6 DAG (5352 genes) and 16 DAG (4701 genes, Fig. 6A, B). The large number of DEGs indicate major developmental changes in the distal region of the seminal roots.

Two lines of evidence indicate similar changes in the transcriptomes of 1RW and UBI::OPRIII-R5 (transformed into 1RS) relative

to 1RS (WT). First, we observed a large number of shared DEGs at 6 DAG (613 genes) and 16 DAG (732 genes) (Fig. 6A, B). In addition, we detected a highly significant regression ($P < 0.001$, $R = 0.76-0.78$) between the $\log_2$ expression ratios in 1RW/1RS and in UBI::OPRIII-R5/1RS for all genes with more than two-fold difference in expression (Fig. 6C, D and the associated source data). These results indicate similar transcriptome changes in 1RW and UBI::OPRIII-R5 relative to 1RS, supporting the hypothesis that the observed changes in 1RW are mainly driven by the increased dosage and expression of the *OPRIII* genes.

To characterize the main pathways affected by the changes in *OPRIII* expression, we carried out an additional Quant-Seq transcriptome analysis comparing 1RW (WT) and mut-OPRIII-B1 (source data of Fig. 6). We performed KEGG analyses (Kyoto Encyclopedia of Genes and Genomes) using the closest rice homologs of the DEGs between the 1RW/1RS and UBI::OPRIII-R5/1RS comparisons at 6 and 16 DAG and for RW/mut-OPRIII-B1 comparisons at 6 and 20 DAG (source data of Fig. 6). These analyses showed three pathways that were significantly enriched in all three comparisons in any combination across the two time points (Fig. 6E). The *P* values for significant enrichment for each pathway (source data of Fig. 6) were calculated using a $2 \times 2$ contingency table comparing the number of hits over the total in the selected list and the population (https://david.ncifcrf.gov/helps/functional_annotation.html#E3). The significant enrichment in the alpha-linolenic acid and glutathione pathways are likely associated with the observed changes in JA (Fig. 4D, E) and ROS (Fig. 3E, Fig. 5C and previous results[14]) in the roots, respectively. An additional pathway significantly enriched at both time points (Fig. 6E) was the phenylpropanoid biosynthetic pathway, which is critical for the establishment of root water barriers[32]. In summary, the three root transcriptome datasets presented in this study represent a valuable resource for functional studies of root genes in wheat.

## Discussion

Altered levels of the phytohormone JA-Ile were shown in different studies to affect root architecture[19,20] and responses to drought[21,22,33] in different plant species. However, natural variation in JA- biosynthesis and signaling, as well as its incidence in root architecture of major crops remains largely unknown. This work shows that differences in *OPRIII* gene dosage underlie the variation between 1RS and 1RW in root architecture, in their ability to access water from deep soil layers, and in their yield potential under limited water condition[13,14]. Knock-outs of single members of this multigene family (Fig. 2A, B) were sufficient to generate changes in seminal root length in 1RW, suggesting that this is a sensitive regulatory point of the JA pathway.

We show here that the understudied genes from the monocot-specific *OPRIII* subfamily encode cytoplasmic and nuclear 12-OXOPHYTODIENOATE REDUCTASE enzymes that regulate a critical step in the synthesis of JA-Ile. Using transcriptome and pharmacological studies, we demonstrate that the effects of the *OPRIII* dosage or expression on root development are mediated by changes in JA-Ile and the downstream JA-signaling pathway in the distal region of the seminal roots. The addition of the JA-biosynthesis inhibitor IBU not only eliminated the differences between 1RS and 1RW in root length, but also in ROS distribution (Fig. 5C).

Our previous results indicate that the JA and ROS pathways may be interconnected, a hypothesis also supported by a previous study in Arabidopsis showing that changes in ROS distribution play an important role in root stem cell maintenance[34]. There is also evidence that ROS, and in particular glutathione, are important components of the root growth regulatory pathway in maize affecting both the meristematic and elongation zones[35]. This connection is supported by our root tip transcriptome studies. Our KEGG analyses comparing the 1RS, 1RW, *OPRIII* mutants and transgenic plants overexpressing *OPRIII* showed

consistent effects on the glutathione pathway, which is known to contribute to the control of redox homeostasis.

Another consistently enriched pathway in the KEGG analyses was the phenylpropanoid biosynthetic pathway, which is critical for the establishment of root water barriers. Autonomous production of phenylpropanoids is required for the establishment of the endodermal Casparian strip as well as for adherence of the suberin matrix to the cell wall of the endodermis[32]. Additional support for the role of the *OPRIII* genes in the regulation of this pathway comes from the increased expression of the rice homologs *Os06G0215500* and *Os06G0215600* in the endo and exodermis of the rice roots (Supplementary Fig. 10)[36].

The rice root spatial heatmap for *OPRIII* genes *Os06G0215500*, *Os06G0215600*, *Os06G0215900* and *Os06G0216300* (http://spatialheatmap.baileyserreslab.org/)[36] also revealed increased expression of the *OPRIII* genes in the apical region, including the quiescent center (Supplementary Fig. 10). This information, combined with the expression of the wheat *OPRIII* genes in the distal part of the seminal roots at different time points (Supplementary Fig. 3) and the developmentally regulated arrest of the RAM in 1RW and UBI::OPRIII transgenic plants, suggest a role of the *OPRIII* genes in the developmental regulation of the RAM in cereal plants.

In Arabidopsis, *PLT* genes have been shown to be dose-dependent master regulators of root development[37], and in rice they are expressed in the root stem cell niche and in the nascent lateral roots[38]. Therefore, it is possible that the observed increase in the expression of the wheat *PLT1*, *PLT3*, and *PLT5* in the lines with increased *OPRIII* dosage or expression (Supplementary Fig. 8) contributed to the arrest of the root meristem and/or to the different distribution of the lateral roots. Since PLT protein gradients are critical for their roles in root development[39], it would be interesting to investigate the effect of changes in *OPRIII* dosage on the spatial distribution of wheat PLT proteins along the roots.

In addition to the basic biological questions that can be investigated by the manipulation of the *OPRIII* genes, this study also points to potential practical applications. One example is the increased root length in the 1RW CRISPR-Cas9 mutants, which provides a path to restore the good performance of the 1RS lines under water stress to the 1WW line (Fig. 1A) with improved breadmaking quality[10,11]. However, the seminal roots of the single gene mutants were still shorter than the original 1RS line (Fig. 2B), suggesting that the *OPRIII* gene dosage may need to be fine-tuned to restore root growth and grain yield potential to the 1RS levels. Finally, the extensive variation detected in the number of functional *OPRIII* genes in the available sequenced wheat genomes (source data of Supplementary Figs. 1 and 2) suggests that natural variation in these genes may have contributed to the adaptation of wheat to different soil environments. The identification of *OPRIII* gene dosage as a sensitive point in the regulation of the JA-biosynthetic pathway provides a target to engineer root architecture in wheat and possibly other cereal crops.

## Methods
### Hahn-1RS and 1RW isogenic lines
The cultivar Hahn, developed by the Centro Internacional de Mejoramiento de Maíz y Trigo (CIMMYT), carries the 1RS.1BL translocation and is referred here as Hahn-1RS. Hahn-1RS was used as recurrent parent for the introgression of two interstitial segments of the 1BS arm from Pavon by homeologous recombination (1WW, PI 672837, Fig. 1A)[12]. The 1WW chromosome was then backcrossed six times into Hahn-1RS and the isogenicity of the BC$_6$F$_2$ sister lines was confirmed using the wheat Illumina 90,000 SNP iSelect array[13]. The Hahn 1WW line was crossed to the original cv. Hahn-1RS to generate a line carrying only the proximal 1BS segment (1WR, PI 672838, for proximal wheat and distal rye segments) and another one carrying only the distal 1BS segment (1RW, PI 672839) (Fig. 1A).

## Generation of diiso 1RS and determination of its effect on root length

An isochromosome 1RS was developed by centric mis-division of the 1RS.1BL translocation in wheat[40]. This translocation originated from the Kavkaz source of the translocation, via cv. Genaro. After self-pollination, a diisosomic 1RS line of Pavon 76 was isolated. For this study, the diiso 1RS of Pavon 76 was crossed, and backcrossed four times to Hahn-1RS with cytological selection for the presence of diiso 1RS in each generation. The $BC_4F_1$ was self-pollinated and individual plants with diiso 1RS as well as telo 1RS were isolated, grown and self-pollinated. As Hahn-1RS has the 1RS.1BL translocation, the homozygous diiso 1RS addition line (2n = 44) has six doses of the rye chromosome 1RS. A cytogenetic study of the progeny of a diiso 1RS plant showed that this extra diiso chromosome is not stable and is lost in approximately 31% of the progeny (source data of Fig. 1).

To determine the 1RS copy number variation (CNV) in 23 progenies of a diiso 1RS plant, we performed six independent DNA extractions for each of the 23 plants, and from 8 Hahn-1RS and 8 Hahn-1RW additional plants as controls. We then used qRT-PCR primers qrt1RS5-F and -R (source data of Supplementary Fig. 3) to determine the dosage of the *OPRIII-R5* gene located in the 1RS chromosome arm. We used the *CO2* gene as an endogenous control for a single copy gene[41]. For each of the 23 progenies and 16 controls, we determined root length in hydroponic tanks as described below, and then calculated a regression between the 1RS CNV and root length using SAS v9.4. To select plants with more than two 1RS chromosomes, we performed *t* tests between the CNV in each recombinant and the 1RS control (two 1RS arms). Root length in plants with more than two 1RS arms were compared with plants carrying the 1RS or 1RW controls. Raw data and statistical analyses for these experiments are available in source data of Fig. 1.

## Hydroponic experiments with mannitol

To simulate osmotic stress, we added D-Mannitol (200 mM) (Sigma-Aldrich, M9647) to the hydroponic solution at 14 DAG[42]. The solution was changed every two days to reduce bacteria proliferation based on the use of mannitol. We measured seminal root and leaf length and dry weight, and also evaluated leaf water status using the relative water content (RWC) method. First, we weighed the second youngest leaf blade (FW) immediately after harvest (two hours after the beginning of the light period), and then we placed it in distilled water for 4 h to evaluate fresh weight at saturation (FWS). We then recorded the dry-weight (DW) after the samples were dried at 60 °C for 5 days. The RWC was calculated based on the following formula: RWC = 100* ((FW-DW)/(FWS-DW))[43].

## Hydroponic experiments with Ibuprofen and MeJa

Seeds were imbibed for 3 days at 4 °C and then sowed on a floating mesh in a 0.5 mM $CaCl_2$ solution at room temperature. Four days later, healthy seedlings were transferred to 350 mL pots (one plant per pot) containing an aerated nutrient solution with the following composition: $Ca(NO_3)_2$ 1.0 mM, KCl 1.5 mM, $KH_2PO_4$ 0.2 mM, $MgSO_4.7H_2O$ 1.0 mM, $CaCl_2$ 1.5 mM, FeEDTA 0.1 mM, $H_3BO_3$ 1 μM, $(NH_4)_6Mo_7O_{24}.4H_2O$ 0.05 μM, $CuSO_4.5H_2O$ 0.5 μM, $ZnSO_4.7H_2O$ 1 μM, $MnSO_4.H_2O$ 1 μM, brought to pH 6.0 ± 0.1 with $Ca(OH)_2$. The solution was renewed three times a week for the duration of the experiment. The experiments were performed in a growth chamber set at 22-23 °C with a photoperiod of 16 h light/8 h dark provided by a fluorescent light source supplemented with incandescent lighting. Photon flux density at the plant level was 150 μmol $m^{-2}$ $s^{-1}$. The length of the second longest seminal root was measured with a ruler four hours after the start of the light period. Pharmacological treatments were imposed by adding either Methyl jasmonate (MeJa; SIGMA: 392707) or Ibuprofen sodium salt (IBU; SIGMA: I1892) from appropriate stock solutions. Nitroblue tetrazolium (NBT) staining was used to visualize ROS.

## Stable transformation

Transgenic Hahn-1RS and 1RW plants were transformed via particle bombardment at UC Berkley. Seeds of these two lines were sown weekly and grown in growth chambers. Immature embryos (IEs) were prepared according to the same specifications as Fielder[44]. They were pre-incubated at 26 °C overnight and used for bombardment. On the day of bombardment, IEs were placed on to DBC3 medium containing mannitol and sorbitol (0.2 M each)[44,45] for osmotic pretreatment. Four hours after treatment with osmoticum, IEs were bombarded as in published study[44] with a few modifications described below.

Two milligrams of 0.6 μm gold particles were coated with 5 μg of a mixture of pAct1IHPT4 and pJIT163-UBI::OPRIII-B1 or pJIT163-UBI-OPRIII-R5 at a 1:2 ratio for the transformation of Hahn-1RS (Supplementary Fig. 4). For the transformation of Hahn-1RW, we used the same amount of gold particles coated with 10 μg of a mixture of pAct1IHPT4, pOsUbi10Cas9 and pENTR/Zeo-H2B-211 at a 1:2:3 ratio (Supplementary Fig. 4). pENTR/Zeo-H2B-211 included two guide RNAs targeted to a conserved region in *OPRIII-D3*, *OPRIII-A2*, *OPRIII-A3*, *OPRIII-B1*, and *OPRIII-B2* (sg51) and *OPRIII-A2*, *OPRIII-A1*, *OPRIII-B1*, *OPRIII-B2*, *OPRIII-D5*, *OPRIII-D2*, *OPRIII-D1*, *OPRIII-R1*, *OPRIII-R5* (sg52, Supplementary Fig. 4). Each particle preparation was resuspended in 85 μL of 100% EtOH and 7.5 μL was spread onto the center of a macrocarrier inside of a macrocarrier holder. The microparticle preps were used for bombardment with a Bio-Rad PDS-1000/He biolistic device (Bio-Rad, Hercules, CA) at 650 psi. Each plate of IEs was bombarded twice per treatment. Sixteen to 18 h post bombardment, IEs were transferred to DBC3 medium and incubated at 26 °C for one week in dim light. Following the resting period, IEs went through 3 rounds of selection via DBC3 media containing 30 mg/L hygromycin B, each round of selection lasting 3 weeks. After the third round of selection, regeneration was initiated using DBC6 media[46] containing 30 mg/L hygromycin B and incubated at 26 °C in high light (90–100 μmol $m^{-2}$ $s^{-1}$) and subcultured every 3 weeks. Once shoots were approximately 0.5–3.0 cm in height, they were transferred to WR rooting media containing 30 mg/L hygromycin B. Plantlets were then transferred to soil once they had enough roots to support transplant to soil.

Initially, Hahn-1RW was co-bombarded with vectors pOsUbi10-Cas9 (Supplementary Fig. 4) and pENTR/Zeo-H2B-211 including guide RNAs Sg51 and Sg52 at UC Berkeley (Supplementary Fig. 4). Later, we performed a second CRISPR screen at the UC Davis Transformation facility using the GRF4–GIF1-CRISPR–Cas9 technology[47]. We designed a single guide RNA (sg= CTCCTCACGCCGTACAAGAT) targeted to a conserved region in *OPRIII-R1*, *OPRIII-D2*, *OPRIII-A3*, *OPRIII-B3*, and *OPRIII-B2*.

Mutations in the Hahn-1RW plants transformed with the CRISPR-Cas9 vectors were screened for mutations in multiple *OPRIII* genes by next-generation sequencing (NGS, MiSeq, Illumina at the UC Davis Genomic Center). Amplicons were obtained using the genotyping method based on custom amplicon sequencing[48] using nonspecific primers to detect mutations in genes with high identity levels (primers are listed in source data of Supplementary Fig. 3). Variants were called for each sample (demultiplexed fastq files) using CRISgo v5 (https://github.com/pinbo/CRISgo). Demultiplexed reads were also mapped to potential targets using BWA v0.7.17 (bwa mem command)[49]. Variants called from CRISgo were validated by visual inspection of the bam files using IGV v2.7.2 (https://software.broadinstitute.org/software/igv/)[50].

In the first CRISPR screen with RNA guides sg51 and sg52 (Supplementary Fig. 4), we identified only one 32-bp frame-shift deletion in the second exon of *OPRIII-B1* (*TraesCS1B02G018700*). The deletion is located between positions 274 and 305 of the coding region, counting from the start codon. This mutation was designated as mut-OPRIII-B1 and encoded a truncated protein where 75 % of the amino acids were altered or eliminated due to a premature stop codon. We segregated out the CRISPR-Cas9 transgene, and selected mut-OPRIII-B1 $T_3$ sister

lines with and without the 32-bp deletion were evaluated in hydroponic experiments.

In the second CRISPR screen with a different sgRNA (CTCCTCACGCCGTACAAGAT), we identified two independent deletions in the rye gene *OPRIII-R1*, which is duplicated in 1RW. Based on the number of reads carrying the deletions, we determined that only one of the copies was edited in each event. The first frame-shift deletion of 7-bp was detected in the first exon (between positions 26 to 32) and resulted in amino acid changes or truncation of 98% of the predicted protein by a premature stop codon. This mutation was designated as mut-OPRIII-R1.7.

The second deletion of 3-bp, designated as mut-OPRIII-R1.3, was also present in the first exon (between positions 30 to 32). This is not a frame-shift deletion, but it affected two adjacent codons and generated a premature stop codon that truncated 97% of the protein.

T$_2$ plants segregating for both deletions and additional nontransgenic sister lines were evaluated for root length in hydroponic experiments. Although the CRISPR transgene was still segregating in this experiment, we did sequence the other *OPRIII* genes targeted by the single guide RNAs (*OPRIII-R1*, *OPRIII-D2*, *OPRIII-A3*, *OPRIII-B3*, *OPRIII-B2*) and did not find any additional mutations in the analyzed plants. In summary all three CRISPR mutants can be considered loss-of-function mutations.

The hexaploid wheat Fielder transgenic plants were generated at the UC Davis Plant Transformation Facility (http://ucdptf.ucdavis.edu/) using the Japan Tobacco (JT) vector pLC41 (hygromycin resistance) and *Agrobacterium*-mediated transformation technology licensed to UC Davis. The wheat *OPRIII-B1* and rye *OPRIII-R5* were cloned in the binary vector pLC41 (Japan Tobacco) downstream of the maize *UBIQUITIN* promoter and upstream of the 3xHA tag and a NOS terminator. Hygromycin was used as a selectable marker for transformation[47]. Plants were grown in hydroponic tanks, and root length measurements were performed every two to three days starting at 7 days after germination (DAG). After the last measurement, leaf samples were collected to extract RNA and DNA to identify the transgenic plants using primers described in source data of Supplementary Fig. 3.

## qRT-PCR of *OPRIII* genes

The expression levels of several *OPRIII* genes were characterized using quantitative reverse transcription PCR (qRT-PCR) using primers described in source data of Supplementary Fig. 3. Hahn-1RS and 1RW plants were grown in hydroponic tanks and the terminal 1 cm of the seminal roots of each plant were collected at different time points (3, 6, 9, 12, and 15 DAG). Roots from 12 plants were pooled per replication to obtain sufficient RNA, and four pools were used as replications for each time point / genotype combination.

RNA samples were extracted using the Spectrum Plant Total RNA Kit (Sigma-Aldrich). First-strand cDNAs were synthesized from 1 μg of total RNA using the High Capacity Reverse Transcription kit (Applied Biosystems). Quantitative PCR was performed using SYBR Green and a 7500 Fast Real-Time PCR system (Applied Biosystems). *ACTIN* was used as an endogenous control. Transcript levels for all genes are expressed as linearized fold-*ACTIN* levels calculated by the formula $2^{(ACTIN\ CT - TARGET\ CT)} \pm$ standard error (SE) of the mean, which indicates the ratio between the initial number of molecules of the target gene and the number of molecules of *ACTIN*.

## Determination of OPRIII enzymatic activity

To overcome the redundancy of *OPRIII* homologs and paralogs, we first amplified *OPRIII-B3* and *OPRIII-B1* from 1RW with primers on the UTR region (OPRIII-B3-F / R and OPRIII-B1-F / R, source data of Supplementary Fig. 3). We then cloned the PCR products into T vector and sequenced them to confirm the presence of the complete coding region. We next added the attB site to the coding region using primer OPRIII-B-attB-F combined with either OPRIII-B3-attB-R or OPRIII-B2-

attB-R (source data of Supplementary Fig. 3). We were not able to clone *OPRIII-B2* and *OPRIII-A2* from the reverse transcription products, so we cloned each exon from genomic DNA using the primers described in source data of Supplementary Fig. 3. Using the genomic clones as templates we obtained the full-length coding region by overlap PCR. The *OPRIII* genes were cloned into vector pDONR207 using the Gateway BP Clonase® enzyme and then transformed into pHIS9, a Gateway compatible destination vector modified from pET28a[51]. The destination plasmids were transformed into *E. Coli*, Rosetta (DE3) for protein expression.

One positive colony per clone was picked into a 10 mL LB liquid medium and cultured overnight at 37 °C. One mL was transferred into 1 liter of LB liquid medium and cultured for ~6 h until OD$_{600}$ 0.6. A stock of isopropyl-β-D-thiogalactoside (IPTG) was added at 1 mM into the cell culture, which was cultivated for 20 h at 16 °C. The cells were collected by centrifugation and stored overnight at −80 °C. The cells were resuspended in 30 mL of lysis buffer (50 mM Tris pH 7.4, 500 mM NaCl 20 mM Imidazole 1% Triton-X100, 1 mM PMSF). The cells were sonicated for 45 min with 5 s sonicate/25 s stop cycles until the sample was clear. The samples were centrifuged at 5,500 g for 10 min at 4°C. The supernatant was centrifuged at 20,000 g for one hour at 4 °C. The supernatant was transferred to a clean 50 mL tube and mixed with 100 μL of Ni-NTA beads Qiagen, Item No. (30210) for 2 h at 4 °C with gentle rotation at 30 rpm. The samples were centrifuged at 500 g for 30 min at 4 °C to collect the pellet. The pellet was washed with 1 mL wash buffer (50 mM Tris pH 7.4, 500 mM NaCl 20 mM Imidazole 1% glycerin) and centrifuged at 500 g for 30 s to remove the supernatant. We repeated the above step twice to remove supernatant contaminations. Pellets were resuspended with 300 μL of elution buffer (50 mM Tris pH 7.4, 500 mM NaCl 250 mM Imidazole 1% glycerin) and kept at 4 °C for 30 min to elute the recombinant protein.

The enzyme activity assay system contained 50 mM PBS, 1 mM NADPH, and 3 μg of the substrate at a 200 μL volume in UV compatible clear 96-well plates (Corning). One μg of recombinant protein was added into the above system to initiate the reaction. OD340 was recorded every minute for 1 h. A boiled mixture of recombinant proteins was used as negative control. A NADPH dilution from 0.25, 0.5, 0.75, 1, and 1.5 mM was used to construct the standard curve. All the samples were analyzed with four replicates.

The 12-oxo-10,15(Z)-phytodienoic acid substrate (CAS No. 85551-10-6) was purchased from Larodan (Item No. 13-1821), whereas the 13-*epi*-12-oxo-phytodienoic acid (CAS No. 71606-07-0) was purchased from Cayman Chemical (Item No. 10195). The synthesis of the 4,5-didehydro jasmonic acid substrate (complete name (+)-3(R),7(S)-4,5-didehydrojasmonic acid, CAS No. 123357-39-1) was reported in a study by Chini et al.[26]. NADPH was purchased from Real-Times, Beijing Biotechnology (Item No. 041939).

## Subcellular localization

The *OPRIII-A2*, *OPRIII-B1*, *OPRIII-B2*, and *OPRIII-B3* genes were cloned from the *T. aestivum* cv. Kern and were integrated into the Gateway vector pDNOR207 using the BP/LR Clonase® reaction (Invitrogen, Life Technologies, Carlsbad, CA, USA). The correct insertion and sequence of the genes was confirmed by Sanger sequencing. For the subcellular localization experiments, the vectors carrying the different *OPRIII* genes were incorporated into pMDC83 (CD3-742, ABRC) to create P35S::OPR-GFP by LR reaction (Invitrogen). All vectors were transformed into *Agrobacterium tumefaciens* GV3101, and then infiltrated into tobacco leaves. After incubation in dim light for 36 h, the samples were observed under a Leica TCS SP8 (Leica Microsystems, Mannheim, Germany). The 35 S::peroxisome-mCherry marker CD3-983[28] was obtained from Arabidopsis Biological Resource center (https://abrc.osu.edu/).

## Jasmonic acid, JA-Ile and OPDA determinations

Hahn-1RS, 1RW and UBI::OPRIII-R5 plants were grown in hydroponic tanks as described above. At 6 DAG, the terminal 1 cm of the seminal roots was collected. To collect 100 mg of fresh tissue, samples from eight plants were collected and pooled for each genotype grown in the same tank. Five tanks were used for collection of independent biological replicates. Samples were flash frozen and provided to the UC Riverside Metabolomics Core Facility, where the 100-mg samples were lyophilized and ground and mixed with 500 μL of 6:3:1 methyl tert-butyl ether (MTBE):methanol:water with the addition of deuterated internal standards. Samples were vortexed 90 min at 4 °C then centrifuged 30 min at 3000 x g and 4 °C, and 200 μL supernatant was transferred to a new 2 mL autosampler vial. Extracts were then dried under nitrogen gas, resuspended in 200 μL methanol, vortexed for 5 min to mix, and transferred to an insert for LC-MS analysis.

A standard curve was prepared by first adding 40 μL of phytohormones standards mix at 10 μg/mL to 760 μL 6:3:1 MTBE:methanol:water with deuterated internal standards, and the subsequent preparation of 2-fold serial dilutions. 200 μL aliquots were dried under nitrogen gas, resuspended in 200 μL methanol, vortexed for 5 min to mix, and transferred to inserts for analysis.

Phytohormone quantitation was performed on a TQ-XS triple quadrupole mass spectrometer (Waters) coupled to an I-class UPLC system (Waters)[52]. Separations were carried out on a T3 C18 column (2.1 × 100 mm, 1.8 μM) (Waters). The mobile phases were (A) water and (B) acetonitrile, both with 0.1% formic acid. The flow rate was 300 μL/min, and the column was held at 45 °C. The injection volume was 2 μL. The gradient was as follows: 0 min, 0.1% B; 1 min, 0.1% B; 6 min, 55% B; 7 min, 100% B; 8 min, 100% B; 8.5 min, 0.1% B; 13 min, 0.1% B. The MS was operated in selected reaction monitoring mode. Source and desolvation temperatures were 150 °C and 600 °C, respectively. Desolvation gas was set to 1100 L/h and cone gas to 150 L/h. Collision gas was set to 0.15 mL/min. All gases were nitrogen except the collision gas, which was argon. The capillary voltage was 1 kV in positive ion mode and 2 kV in negative ion mode. A quality control sample, generated by pooling equal aliquots of each sample, was analyzed periodically to monitor system stability and performance. Samples were analyzed in random order.

Skyline-Daily software (latest version, MacCoss Lab, Seattle, WA) was used to detect and integrate peak areas. Exported peak areas was used to calculate linear regression of analytical standards used for quantification. JA was normalized to isotopically labeled internal standard. The corresponding linear regression equation was used for quantification (ng/ml) for each analyte, which was then adjusted for precise weight of leaf tissue for each sample (ng/g). For JA-Ile and OPDA, for which there are not internal standards for, peak areas were used for final statistical analyses.

## RNA-seq of 1RW, 1RS and UBI::OPRIII-R5

The terminal 1 cm of the three seminal roots of each plant were collected at 6 and 16 DAG. Roots from 12 plants were pooled per replication to obtain sufficient RNA, and four pools were used as replications for each time point/ genotype combination. RNA samples were extracted using the Spectrum Plant Total RNA Kit (Sigma-Aldrich).

Messenger RNA was purified from total RNA using poly-T oligo-attached magnetic beads. After fragmentation, the first-strand cDNA was synthesized using random hexamer primers, followed by the second strand cDNA synthesis using dTTP for a non-directional library. The library for transcriptome sequencing was ready after end repair, A-tailing, adapter ligation, size selection, amplification, and purification. The library was checked with Qubit and real-time PCR for quantification and bioanalyzer for size distribution detection. The quantified libraries were pooled and sequenced on Illumina platforms. The clustering of the index-coded samples was performed according to the manufacturer's instructions (Novogene). After cluster generation, the library preparations were sequenced on an Illumina platform and paired-end reads were generated. The number of reads per sample and different quality and mapping statistics are described in source data of Fig. 6.

Reads were mapped to the Chinese Spring Genome RefSeq v1.0[25] combined with the 1RS arm from cultivar Aikang58 (AK58)[53], allowing a maximum of 1 SNP. Reads were mapped using the splicing aware STAR aligner from the Lexogen pipeline. Reads mapping to more than one location were distributed equally among the identical targets. Expression values were calculated using the trimmed mean of M-values normalization method (TMM, source data of Fig. 6)[54]. The sequence of the 1RS.1BL translocation in AK58 is available only as a preprint and no final gene names have been published, so we provide a table with the different names and genome coordinates to facilitate future cross-reference (source data of Fig. 6)[53].

## Quant-seq for 1RW and mut-OPRIII-B1 in RW

We performed the RNA-extraction as described above (RNA-seq). Samples were collected at 6 and 20 DAG. The second collection was done at 20 DAG since the differences in root length were more significant at this time point. A later collection time relative to the RNA-seq experiments was necessary because the effect of the single gene mutation in mut-OPRIII-B1 on root length is expected to be smaller than the duplication of multiple *OPRIII* genes in 1RW or the constitutive expression in UBI::OPRIII-R5.

Sequences from the 16 samples (2 genotypes × 2 developmental stages × 4 biological replicates) were generated using Hi-seq (100-bp reads not paired) at the UC Davis Genome Center. The number of Quant-Seq reds per sample and different quality and mapping statistics are described in source data of Fig. 6. We processed the raw reads using DOE JGI BBTools (https://sourceforge.net/projects/bbmap/) program bbduk.sh to remove Illumina adapter contamination and low-quality reads (forcetrimleft = 21 qtrim = r trimq = 10). Reads were mapped and TMM values were calculated as described above (RNA-seq, source data of Fig. 6).

All successfully mapped reads were subjected to differential expression analysis using the DESeq2 R package[55] in comparisons between 1 RW *vs*. 1RS and UBI::OPRIII-R5 *vs*.1RS, both at 6 and 16 DAG and 1RW *vs*. mut-OPRIII-B1 at 6 and 20 DAG. Only transcripts with CPM > 0.5 in at least 2 samples were included for the last step of the analyses. Genes with an adjusted *P* value based on a false discovery rate (FDR < 0.05) were considered significant DEGs.

To explore the similarity between the changes in the root transcriptomes of UBI::OPRIII-R5 and 1RW relative to 1RS, we performed a regression analysis between the log$_2$ fold changes in these two comparisons. We first eliminated all genes with 0 counts in all four reps in any of the genotypes (to avoid ratios with 0 as denominator and logs of 0), and then calculated the average TMM for each gene at 6 and 16 DAG for the three genotypes. We determined the log$_2$ of the ratios between the averages in 1RW / 1RS and UBI::OPRIII-R5 / 1RS for 6 and 16 DAG separately and for each day retained only those genes showing changes in expression larger than two-fold for the two ratios (log2 ratio < −1 or > +1). We then performed a regression analysis and plotted the results using PROC REG in SAS 9.4.

A Principal Component Analysis (PCA) of the 24 RNA-seq samples (3 genotypes x 2 time-points x 4 biological replications) was carried out with pcaExplorer using 8,862 differentially expressed genes[56]. The extraction and visualization of the overlapping DEGs between the comparisons of 1RS vs 1RW and 1RS vs UBI::RS6.4 at 6 and 16 DAG was performed with Venny (https://bioinfogp.cnb.csic.es/tools/venny/index.html).

The DEGs obtained from the comparisons between 1RW *vs*. 1RS, UBI::OPRIII-R5 *vs*. 1RS and 1RW *vs*. mut-OPRIII-B1 were subjected to separate KEGG analyses at 6 DAG and 16 or 20 DAG. The analyses were

carried out using DAVID[57,58]. Enriched pathways were considered significant using $P < 0.05$. Genes were classified as up (ratio >1) or down-regulated (ratio <1) using the ratio between the higher dosage-expression OPRIII genotypes over the lower one (1RW/1RS; UBI::OPRIII-R5/ 1RS, and 1RW/mut-OPRIII-B1).

## Reporting summary

Further information on research design is available in the Nature Portfolio Reporting Summary linked to this article.

## Data availability

RNA-seq data for Hahn-1RS, Hahn-1RW, and Hahn-UBI::OPRIII-R5 were deposited in GenBank under BioProject PRJNA819072, PRJNA819073, and PRJNA819075, respectively. Quant-seq data for Hahn-1RW and mut-OPRIII-B1 were deposited in GenBank under BioProject PRJNA847262 and PRJNA847590, respectively. Genome sequences for the wheat Chinese Spring Genome RefSeq v1.0 and the different wheat genomes sequenced in the wheat pan-genome project can be accessed at GrainGenes [https://wheat.pw.usda.gov/GG3/genome_browser]. The genetic stocks used in this study have been deposited in the United States Department of Agriculture National Small Grains Collection [https://www.ars.usda.gov/pacific-west-area/aberdeen-id/small-grains-and-potato-germplasm-research/docs/national-small-grains-collection/] as PI 672837, PI 672838, and PI 672839. Source data are provided with this paper.

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

## Acknowledgements

J.D. and T.F. acknowledge support from USA-Israel BARD grants US-5191-19C and US-5515-22C. J.D. also acknowledges support from the USDA National Institute of Food and Agriculture the Agriculture and Food Research Initiative Competitive Grant 2022-68013-36439 (WheatCAP), and from the Howard Hughes Medical Institute. G.G. acknowledges support from Vaadia-BARD fellowship number FI-585-2019. G.E.S.-M. acknowledges support from CONICET and the ANPCYT (PICT 2018-02159) Argentina. J.G. acknowledges support from the National Natural Science Foundation of China Grant 31972350. We thank the UC Riverside Metabolomics Core Facility for their help with the Jasmonic Acid determinations and the UC Davis transformation facility for the Fielder transgenic plants. We also thank Yanpeng Wang for his help with the transformation vectors; Rudi Appels for his help with the 1RS gene names in AK58; and Neelima Sinha and Kristina Zumstein for their assistance with the amplicon sequencing to detect induced CRISPR-Cas9 mutations.

## Author contributions

G.G. performed most of the experimental work and data analysis, supervised H.W., and wrote the first version of the manuscript. J.Z., G.S.M., J.M., G.F.B., and J.D. contributed data analyses. H.W., J.Z., T.H., contributed to the experimental work. J.M., L.D.G. and G.S.M. performed the Ibuprofen and MeJa experiments and evaluated the nodal roots. B.S., M.J.C. and J.T. contributed the Hahn transgenic plants. H.K. and K.D. performed the J.A. determinations. G.L.Z. and J.Y.G. determined the enzymatic activity of the OPRIII proteins. A.L. contributed the diiso-1RS and other valuable genetic resources, and MH synthesized and provided the 4,5-ddhJA. J.D., T.F., and G.S.M. wrote the grant proposals that supported this work. All authors reviewed the manuscript. J.D. and G.S.M. initiated the project and supervised students and postdocs. J.D. contributed to the statistical analyses and was responsible for the final manuscript.

## Competing interests

The authors declare no competing interests.
