## [Peer Review File · Nature Communications]

Dosage differences in 12-OXOPHYTODIENOATE REDUCTASE genes modulate wheat root growthReviewers' Comments:

Reviewer #1:

Remarks to the Author:

Greater understanding about the genetic regulation of root architecture in crops such as wheat is vital for ongoing food security efforts. The paper (Gabay et al.) builds on previous papers (Howell et al., 2019; Gabay et al., 2021) where it has been shown that 1RW wheat plants encoding an engineered chromosome have slow growing roots as compared to the original 1RS rye introgression line. Exome sequencing of 1RW chromosome and radiation induced deletion of the distal 1BS insertion and 3.3 Mb of the adjacent 1RS duplicated region (Gabay et al., 2021) revealed the importance of gene dosage in conferring a shorter root phenotype in 1RW lines. Gabay et al., 2021 further identified 13 genes which may be involved in modifying root architecture in 1RW. Of these, 4 genes encoded for 12-OXOPHYTODIENOATE REDUCTASE (OPR) which are involved in JA biosynthesis.

In the submitted manuscript, the authors have demonstrated that the short root phenotypes of 1RW line is indeed due to a higher gene dosage of OPR. The authors have confirmed their hypothesis by showing that mutations in OPRs lead to longer root phenotypes and overexpression of OPR leads to shorter root phenotypes. The authors have also measured JA content in their lines and established that high JA accumulation leads to shorter root phenotypes. The authors have well supported their claims through a series of experiments (e.g. generation of stable wheat lines, quantitation of hormone and expression data). The conclusions are (largely) backed up by their experimental evidence (see comments below).

Despite the solid quality of results presented, the overall message of the paper lacks novelty. For example, there is ample evidence in the literature in Arabidopsis and also crops like rice which demonstrates high JA concentrations or enhanced JA signalling leads to root growth inhibition, versus a decrease in JA content promoting root growth. Secondly, much of the work has been previously reported (e.g. Gabay et al., 2021 identified 13 genes involved in modifying root architecture in 1RW, 4 of which encoded OPR genes). Third, rather surprisingly, the physiological (and agronomic) impact of varying OPR gene dosage was not investigated using the new wheat genetic and transgenic lines generated for this study. Fourth, the impact on wheat root classes other than seminal and lateral roots (e.g. tillering roots) and on overall root system architecture at different life cycle stages remains unclear. Therefore, the manuscript does not provide a fundamental advance in the field and is better suited to a specialist journal such as TAG.

Other comments:

1. The authors have shown that 1RW lines and UBI::OPRIII-R5 with more JA are highly sensitive to exogenous JA concentrations as compared to 1RS lines with lesser endogenous JA. This is difficult to interpret as plants with high endogenous JA, usually show less sensitivity to exogenous JA due to the presence of saturated levels of endogenous JA. Furthermore, why are 1RS lines completely insensitive to MeJA application? The authors should check whether the JA concentrations used were ideal to study JA dependent root inhibition phenotype.
2. The authors efforts to specifically block JA biosynthesis using ibuprofen should be cross-validated by performing ibuprofen + meJA controls.
3. The authors have just focussed on JA biosynthesis. They should show the effect of enhanced JA biosynthesis on JA signalling genes and other downstream targets in their transcriptome dataset.
4. The dosage of OPRIII-R5 on 1RS chromosome was determined indirectly by qRT-PCR. As the whole hypothesis revolves around dosage of OPRIII genomic in situ hybridizations using rye chromatin should be performed and precisely fluorescent in situ hybridizations should be performed on same

cells parallelly using gene specific probes.

Reviewer #2:

Remarks to the Author:

Understanding the genetic bases of root architecture in bread wheat is thought to be essential to further improve crop production. Previously, Howell et al 2019 and Gabay et al 2021 have shown that the introgression of rye chromosome one (1RS) into bread wheat confers higher root biomass and a yield advantage under drought stress. Here, Gabay et al identify the underlying genes impacting the root length phenotype. A cluster of jasmonate (JA) biosynthesis genes called OPRIII, impinge on root length, JA and ROS levels depending on their dosage. Low OPRIII copy number resulted in lower OPRIII dosage (or activity) and consequent longer roots/ reduced hormone and ROS levels, while higher OPRIII expression led to shorter roots, higher JA and ROS levels. Transcriptomics and network analyses were in line with these findings. Overall, the authors conclude that OPRIII can be used to modulate wheat root architecture. Findings are exciting and of broad general interest, the data are rigorous and the manuscript is well written and well presented.

MAJOR POINTS

1. The authors should perform drought assays to test if the modulation of root JA levels confers drought tolerance in wheat as expected from model plants. This can be done at the seedling stage in soil (eg. Kim et al 2017 Nat Plants) or hydroponic conditions with the use of eg. mannitol. Both positive or negative findings would greatly strengthen the impact of this work
2. OPDA and JA are the precursors of bioactive JA-Ile, thus not active. Quantifications are provided only for JA (Fig.4). These should be complemented with OPDA and JA-Ile levels in mol/g FW and relative Limit of Quantification values for each compound.

MINOR POINTS

- Fig.5C. Is there a specific reason for root tips facing upwards? If not, I would recommend to orient the meristems downwards for consistency with all other figures in the paper.

Reviewer #3:

Remarks to the Author:

This paper is an excellent contribution to our understanding of root growth regulation, supported by strong evidence from a series of experiments that cover multiple ways of demonstrating the identity and function of the identified genes that are responsible for the observed phenotypes. Furthermore, there are important practical implications of the findings for breeding improved wheat varieties for water-limited environments. I would recommend publication.

This paper extends the research reported by the lead author and others in a 2021 paper (cited), which reported the discovery that gene dosage effects explained the reduction in root growth observed in wheat lines containing a recombinant 1BS.1BL chromosomal translocation from rye that contains an interstitial wheat introgression in the distal region of the translocated rye chromosome (1RW). This report is a continuation of an elegant and painstakingly careful pathway of discovery into the genetic basis of this diminished root growth phenotype, eliminating the most obvious hypothesised causes (loss of 1RS rye genes or added wheat 1BS genes), and concluding that the 1RW introgression was actually quite complex structurally, with multiple gene duplications, causing the observed gene dosage effect. The previous paper identified 14 high confidence candidate genes in the chromosomal region that remained after radiation-induced deletion of two of the duplicated rye segments and two wheat segments in the distal region, leaving 2 duplicated rye segments responsible for the dosage effect. From that list, OPRIII was highlighted as a likely candidate. The current paper identifies the duplicated

gene within that region responsible for the root growth inhibition of 1RW compared with the original 1RS, which through a combination of approaches was confirmed to be the OPRIII gene family.

In addition, experiments described a functional characterisation of expressed OPRIII genes via NADPH consumption assays using OPDA (jasmonate precursor) as substrate. The results were consistent with oxophytodienoate reductase activity.

A series of experiments were conducted to confirm the role of JA in modulating the reduced root growth of lines with increased OPRIII gene number by treating plants with ibuprofen (IBU), which inhibits jasmonyl-isoleucine synthesis, the active form of the phytohormone. Results supported the role of JA by nearly recovering the long root phenotype in the IBU-treated 1RW line, which contains duplicated OPRIII genes. Levels of JA were also elevated in root tips of 1RW compared with 1RS (two OPRIII genes from rye), and were further elevated in the 1RW lines that overexpressed the rye OPRIII (R5).

All experiments were repeated and there was appropriate and thorough statistical analysis throughout.

Minor comments:

- P4L15 and L38: Authors point out that ROS play important roles in the meristem, although the effect on root growth was also likely due to altered cell expansion in the elongation zone. There is evidence that ROS, and in particular glutathione, are important components of the root growth regulatory pathway in maize (Kang et al. 2022 <https://doi.org/10.3390/antiox11050820>). Unless authors are certain that the site of action is exclusively in the meristem, mention of this additional role of ROS would help make the story more complete.
- Authors showed in the 2021 paper that the 1BS duplication had a smaller effect on root growth than the 1RS duplication. It would be helpful if authors explained why therefore the OPRIII-B1 gene was chosen for editing, rather than OPRIII-R5, which was not edited, but used in the overexpression lines (along with B1 as well).
- Use of the ubiquitin promoter, which achieved constitutive over-expression of OPRIII genes including in leaves (P3,L25), could have resulted in poor shoot growth, which could have affected root growth towards the end of the experiment as roots shifted from drawing on seed reserves to receiving photoassimilate from the shoot. Hence the gene effect in O/E lines could have been due to a shoot effect rather than or in addition to a direct effect on roots. This seems an unlikely explanation of the short root phenotype, but perhaps some note on general appearance of the shoots would be warranted.
- In the Introduction it was mentioned that introgression of the 1BS genes into the 1RS rye translocation was done to improve grain quality (e.g. dough strength), but this also resulted in reduced root growth, or at least loss of the improved root growth seen in the intact 1RS. There should be a comment on how it might be possible to preserve the improved quality of the recombinant 1RS translocation while eliminating the duplicated OPRIII genes responsible for deleterious root growth. In the paper it is not made clear if the parental line used in this study was one described by Lukaszewski (2000) that included the wheat glutenin/gliadin genes rather than the line that removed the rye secalin gene to improve quality. Either way, knowledge of the identity and location of the OPRIII duplications should allow targeted selections to maintain root growth and grain quality in parallel. A comment would be warranted.
- P3,L36 Authors should consider wording that takes into account that NBT staining is not specific for superoxide, as NBT apparently can be reduced by other enzymatic reductases present in the tissue. However, it unlikely that a different result would be obtained or that conclusions would be altered by using a different method of quantification.
- Fig. 5: the ordering of Figures, or the lettering order for B and C needs checking.
- A possible point of confusion for readers is why roots of 1RS lines grow faster (or at least biomass is greater) than roots of plants without the rye translocation, with the original 1BS chromosome. Is this because the intact 1RS chromosome contains only two OPRIII genes (R1 and R5; Fig. 1E), whereas the 1BS chromosome contains four (B1, B2, B3, B9), hence greater dosage, even though the 1RS

genes had a stronger effect on root growth than the 1BS genes (Gabay et al 2021)?

- The results show early effects of JA accumulation (via OPRIII activity) on seminal root growth that decay over short periods of time. Sustained root growth in seminal and crown roots must occur in the field and soil-based pot experiments that have shown greater root biomass or root depth of 1RS lines compared with lines without the rye translocation. It is not clear how to connect these seedling results to those observed on more mature root systems. A comment is warranted.
- The authors state at the conclusion of the paper that manipulating OPRIII gene dosage could be a way to engineer improved root systems. However, this study focused on reducing the effects of extraneous dosage responsible for reduced root growth, and targeted deletions (or IBU treatment) were effective only in returning root growth back to wildtype (1RS) rates. Could authors speculate on how further increases in root growth rate (not reductions) would be possible through breeding, either in a 1RS background, or in lines without the rye translocation and only containing wheat OPRIII genes? It seems likely that total elimination of root-specific JA production would go too far, resulting in aberrant control of ROS homeostasis in the root growth zone (and also perhaps loss of beneficial secondary effects of JA-mediated protection of roots against root herbivory and soil pathogens).

Eric Ober

ANSWERS TO REVIEWERS

Nature Communications manuscript NCOMMS-22-25946-T Gabay et al

Reviewer #1:

Greater understanding about the genetic regulation of root architecture in crops such as wheat is vital for ongoing food security efforts. The paper (Gabay et al.) builds on previous papers (Howell et al., 2019; Gabay et al., 2021) where it has been shown that 1RW wheat plants encoding an engineered chromosome have slow growing roots as compared to the original 1RS rye introgression line. Exome sequencing of 1RW chromosome and radiation induced deletion of the distal 1BS insertion and 3.3 Mb of the adjacent 1RS duplicated region (Gabay et al., 2021) revealed the importance of gene dosage in conferring a shorter root phenotype in 1RW lines. Gabay et al., 2021 further identified 13 genes which may be involved in modifying root architecture in 1RW. Of these, 4 genes encoded for 12-OXOPHYTODIENOATE REDUCTASE (OPR) which are involved in JA biosynthesis. In the submitted manuscript, the authors have demonstrated that the short root phenotypes of 1RW line is indeed due to a higher gene dosage of OPR. The authors have confirmed their hypothesis by showing that mutations in OPRs lead to longer root phenotypes and overexpression of OPR leads to shorter root phenotypes. The authors have also measured JA content in their lines and established that high JA accumulation leads to shorter root phenotypes. The authors have well supported their claims through a series of experiments (e.g. generation of stable wheat lines, quantitation of hormone and expression data). The conclusions are (largely) backed by up by their experimental evidence (see comments below).

1. Despite the solid quality of results presented, the overall message of the paper lacks novelty. For example, there is ample evidence in the literature in Arabidopsis and also crops like rice which demonstrates high JA concentrations or enhanced JA signalling leads to root growth inhibition, versus a decrease in JA content promoting root growth

Authors answer: We understand that the role of JA on root development is well known. What is novel in our study is the discovery of a regulatory point in the biosynthetic JA process in the root meristematic region that is subject to regulation by *OPRIII* dosage. This is not only novel from a basic point of view for all plant species, but it is a discovery with important implications for the improvement of a crop species relevant for global food security.

The *OPRIII* subfamily is monocot-specific and is not present in Arabidopsis, so the extensive Arabidopsis literature is not directly applicable to this divergent subfamily that is highly expanded in the grasses and that has received, so far, limited attention. In addition, we identify a new cluster of *OPRIII* genes that is exclusive to the Triticeae and that is absent in rice. We show that the genes from the new locus are more related to each other than to the *OPRIII* located in homoeologous group 7, which are colinear to the rice *OPRIII* genes. Finally, we describe previously unreported natural variation in *OPRIII* copy number among different wheat sequenced genomes and demonstrate that root length can be modulated by changes in a single copy of this multigene family using three independent CRISPR mutants. These results confirm that the *OPRIII* genes represent a sensitive point in the regulations of the JA-biosynthetic pathway, a novel insight with practical biotechnological implications.

We tried to make these ideas clear in the first paragraph of the new Discussion section: “Altered levels of the phytohormone JA-Ile were previously shown to affect root architecture and responses to drought in different plant species^{19, 20, 21, 22}. However, natural variation in JA- biosynthesis and signaling, as well as its incidence in root architecture of major crops remains largely unknown. This work shows that differences in *OPRIII* gene dosage underlie previously described variation between 1RS and 1RW in root architecture, in their ability to access water from deep soil layers, and in their yield potential under limited water condition^{13, 14}. Knock-outs of single members of this multigene family (Fig. 2A-B) were sufficient to generate changes in seminal root length in 1RW, suggesting that this is a sensitive regulatory point of the JA pathway.”

This work also presents the first positional cloning and validation of a drought-related QTL in wheat, and the first positional cloning from an alien introgressed segment in wheat achieved by homoeologous recombination induced by the *ph1b* mutant.

2. Secondly, much of the work has been previously reported (e.g. Gabay et al., 2021 identified 13 genes involved in modifying root architecture in 1RW, 4 of which encoded OPR genes).

Authors answer: The previous paper describes the complex structural changes that occur in the 1RW line and used a radiation mutant to delimit a region including 38 potential candidate genes. The 2021 paper does not identify the causal gene of the drought tolerance QTL, does not provide functional validation by CRISPR and over-expression, and does not demonstrate that the mechanism is mediated by changes in the JA-biosynthetic and signaling pathway. All these important steps are described for the first time in the current study. We also present three previously unpublished RNA-seq and Quant-Seq roots data sets that represent a useful tool for root functional studies in wheat. These transcriptome experiments demonstrate an extensive developmental change and provide evidence of the main pathways altered by the changes in *OPR3* dosage or expression. We also demonstrate in this paper that the wheat *OPR3* genes encode cytoplasmic localized active enzymes, and validate their enzymatic activities in three different substrates. We also demonstrate that the effect of the *OPR3* genes on root elongation and development is mediated by a JA-dependent mechanism by directly measuring JA, JA-Ile and OPDA in the root tips and by using a JA-biosynthesis inhibitor. None of these results are reported in Gabay et al. 2021.

The results reported here represent several years of work by a team of 19 researchers from 10 different institutions in five different countries. It required the transformation of a recalcitrant wheat genotype (Hahn), the characterization of a large multigene family in an hexaploid species and multiple rounds of CRISPR in a species that is not easy to transform, and the synthesis of an enzyme substrate that cannot be purchased commercially. We think that the statement that most of the work in this study has been previously reported is simply incorrect.

We tried to clarify at the end of the introduction, what was previously known and what is new in this manuscript: “This radiation mutant further delimited a candidate gene region including 38 wheat and rye duplicated orthologs, but did not identify the causal gene. In this study, we present the functional validation of the genes responsible for the differences in root development using CRISPR-Cas9 induced mutants and transgenic lines. We also demonstrate that these genes encode cytoplasmic functional enzymes involved in the biosynthesis of JA-Ile, and that a JA biosynthetic inhibitor can abolish the differences between 1RS and 1RW in root architecture. Finally, we present high-throughput transcriptomic data for the isogenic and transgenic lines that reveal the pathways affected by the differences in dosage or expression levels of these genes.”

3. Third, rather surprisingly, the physiological (and agronomic) impact of varying OPR gene dosage was not investigated using the new wheat genetic and transgenic lines generated for this study. Fourth, the impact on wheat root classes other than seminal and lateral roots (e.g. tillering roots) and on overall root system architecture at different life cycle stages remains unclear.

Authors answer: This is not a reverse genetics study where one alters one gene and then needs to demonstrate its effect and utility. This is a forward genetics study where we started from a

significant and large QTL for grain yield under water limited conditions. We FIRST characterized the physiological and agronomic effects of this QTL in extensive field experiment over multiple years and multiple locations. So, the work the reviewer indicates as missing has already been performed and published (Howell et al, 2014 and 2019).

To make this clear, we re-wrote the introduction including a detailed description of the near isogenic nature of the 1RS and 1RW genetic stocks used in the field experiments, and the results of the different field experiments that demonstrate the impact of the cloned QTL in an agricultural context: “We introgressed the engineered 1WW chromosome into the 1RS.1BL cv. Hahn by six backcrosses, and then generated three lines with either the complete 1RS (1RS), the proximal 1BS introgression removing the rye secalins (1WR), or the distal 1BS introgression restoring the *Gli-B1* / *Glu-B3* locus (1RW, Fig. 1A). Using the wheat Illumina 90K SNP iSelect array, we demonstrated that the 1RS and 1RW lines were 99.3% identical ¹³.

In large physiological and agronomic field experiments replicated over four years (2008-2012), in two locations, and under both full-irrigation and terminal drought, lines carrying the complete 1RS arm showed significantly higher grain yield relative to 1RW (up to 40% higher under terminal water stress) ¹³. The 1RS lines showed higher carbon isotope discrimination and increased stomatal conductance, indicating improved access to residual water in the soil. Increased water access was also evident in replicated split-plot field experiments where 1RS plants showed improved water indexes and absence of rolled leaves and dry leaf tips that were evident some years in the adjacent 1RW plots ¹³.

In a subsequent field study, we excavated three ~2 m deep trenches cutting perpendicular across the middle of the plots in three blocks including the different genotypes and took horizontal soil core samples from the center of each plot at 20 cm intervals ¹⁴. These data confirmed that the 1RS lines have a higher root density, with roots that reach deeper in the soil and can access more water than the 1RW lines ¹⁴”.

The previous results show that we have done a detailed physiological and agronomic characterization of the 1RS and 1RW isogenic lines carrying different dosages of the *OPR11* genes. This is more relevant than the proposed comparisons of transgenic lines overexpressing the *OPR11* genes or the CRISPR mutant lines. In addition to the complex regulations for performing transgenic field experiments, the results would be more difficult to interpret due to ectopic expression in the first case and unintended effects of the transformation in the second one. The comparison of the isogenic 1RS and 1RW lines in the field, together with the demonstration in this study that the *OPR11* duplications are the cause of most of the differences between 1RS and 1RW, represented a cleaner alternative for the physiological and agronomic field studies.

We hope that the addition of the abovementioned description of previous results in the introduction will help readers understand the extensive agronomic and physiological characterization that was done before attempting the cloning of the causal genes.

In the revised version, we have also introduced new data on the elongation differences in nodal roots between 1RS and 1RW, which mimic the results found for seminal roots. Therefore, both the seminal and nodal root growth patterns are consistent with the pattern of root growth displayed by these two lines in the field.

4. Therefore, the manuscript does not provide a fundamental advance in the field and is

better suited to a specialist journal such as TAG.

Authors answer: Many of us are editors in journals, including the senior author, who has served as editor in TAG for 26 years. We are certain that this manuscript does not belong to TAG. This is supported by the fact that the manuscript was sent for review in both the Science and in Nature Communications submission (most manuscripts are not sent for review), where they received two positive reviews out of three. The dissenting voice has been the one from Reviewer #1, who submitted the same review to both journals. We think we have addressed properly the three main criticisms from Reviewer #1 in our previous responses.

Other comments:

5. *The authors have shown that 1RW lines and UBI::OPRIII-R5 with more JA are highly sensitive to exogenous JA concentrations as compared to 1RS lines with lesser endogenous JA. This is difficult to interpret as plants with high endogenous JA, usually show less sensitivity to exogenous JA due to the presence of saturated levels of endogenous JA. Furthermore, why are 1RS lines completely insensitive to MeJA application? The authors should check whether the JA concentrations used were ideal to study JA dependent root inhibition phenotype.*

Authors answer: This statement refers to an experiment that was not present in the Nature Communication manuscript. This experiment was in the original Science version, but was taken out to accommodate the above suggestion of Reviewer #1. Perhaps Reviewer #1 did not read thoroughly the new version, and only copied and pasted the previous review to Science.

6. *The authors efforts to specifically block JA biosynthesis using ibuprofen should be cross-validated by performing ibuprofen + meJA controls.*

Authors answer: We thank Reviewer #1 for this suggestion. We performed the requested experiment and included the results in the new Fig. 5D and Data S16. We described the new results as follows: “We also analyzed the effect of combined levels of IBU-5 μM and MeJA-40 μM applied from 6 DAG on 1RS and 1RW root length. Plants treated only with IBU showed similar results to those in the previous experiment, whereas those treated only with MeJA showed inhibited root growth (Fig. 5D). The MeJA treatment alone reduced but not eliminated the differences between 1RS and 1RW genotypes. At 20 DAG, the differences between genotypes were significant ($P < 0.01$ Data S16) in the MeJA treatment but not significant in the combined MeJA-IBU treatment (Fig. 5D, Data S16). A three-way ANOVA at 20 DAG showed significant differences for the three main factors (Genotype, MeJA and IBU) and the three two-way interactions (Data S16). The significant genotype x MeJA ($P < 0.001$) and genotype x IBU ($P < 0.05$) interactions are consistent with the hypothesis that the root length differences between 1RS and 1RW are associated with changes in the JA biosynthetic / signalling pathway.”

7. *The authors have just focussed on JA biosynthesis. They should show the effect of enhanced JA biosynthesis on JA signalling genes and other downstream targets in their transcriptome dataset.*

Authors answer: We think this is a good suggestion. We characterized the expression of the main JA-signaling genes and JA-responsive genes in response to the changes in *OPRIII* dosage (1RW) and expression (UBI::OPRIII-R5) relative to 1RS and added the results in Extended Data Fig. S7 and Data S14. We added in the “Results” section: “To explore the effect of the increases in JA-Ile on wheat genes known to be involved in the JA-signaling pathway (*COII*, *JAZ1*, *JAM3* and *MYC2*, Extended Data Fig. S7) ³⁰, we compared the root transcriptome data from 1RW and UBI::OPRIII-

R5 with 1RS (Data S4). For each gene, we analyzed simultaneously the three homoeologs in a factorial ANOVA including genotypes and homoeologs as factors and RNA-seq samples as replications. No significant interactions between homeologs and genotypes were detected, indicating similar responses among homoeologs. All four genes showed significant differences in expression between 1RS and the genotypes with increased JA-Ile (Extended Data Fig. S7, and Data S14). We also observed a significant increase in the transcripts of the JA-regulated genes *PLETHORA 1 (PLT1)*, *PLT3* and *PLT5* at 16 DAG but not at 6 DAG (Data S14). These results support a role of the *OPRIII* genes on the JA-Ile signaling pathway, which is known to be involved in the regulation of root architecture³⁰.”

In the Discussion we added that the role of the wheat *OPRIII* genes in the developmentally regulated arrest of the RAM in 1RW and UBI::OPRIII transgenic plants was likely mediated by changes in JA-regulated genes: “In *Arabidopsis*, *PLT* genes have been shown to be dose-dependent master regulators of root development³⁶, and in rice they are expressed in the root stem cell niche and in the nascent lateral roots³⁷. Therefore, it is possible that the observed increase in the expression of the wheat *PLT1*, *PLT3*, and *PLT5* in the lines with increased *OPRIII* dosage or expression (Extended Data Fig. S7) contributed to the arrest of the root meristem and/or to the different distribution of the lateral roots. Since PLT protein gradients are critical for their roles in root development³⁸, it would be interesting to investigate the effect of changes in *OPRIII* dosage on the spatial distribution of wheat PLT proteins along the roots.”

8. The dosage of *OPRIII-R5* on 1RS chromosome was determined indirectly by qRT-PCR. As the whole hypothesis revolves around dosage of *OPRIII* genomic *in situ* hybridizations using rye chromatin should be performed and precisely fluorescent *in situ* hybridizations should be performed on same cells parallelly using gene specific probes.

Authors answer: This statement is inaccurate. In addition to the qRT-PCR experiments, gene dosage was determined from read counts in two exome capture experiments using two different capture platforms, as described in Gabay et al. 2021 (data in the supplementary files). We analyzed copy number variation in all genes located in the duplicated region and in the flanking genes so the dosage estimates are highly replicated. We also provided a clear explanation of the origin of the duplication in Gabay et al. 2021 based on an inversion detected between the distal 1BS and 1RS regions.

New information added to the revised manuscript

We took advantage of the recently published expression atlas of the rice roots, to determine the spatial localization of the expression of the four homologous *OPRIII* genes in rice roots. These results indicated a localization of the *OPRIII* in the endo and exodermis, which is consistent with the KEGG results indicating an effect on the phenylpropanoid pathway and in the apical root region and quiescent center, which in turn is consistent with the effect of the increased dosage of the *OPRIII* genes on the developmental arrest of the RAM. We summarized these results in a heat-map of the rice homologs in the roots in Extended Data Fig. S9. We added to the discussion: “Additional support for the role of the *OPRIII* genes in the regulation of the phenylpropanoid pathway comes from the increased expression of the rice homologs *Os06G0215500* and *Os06G0215600* in the endo and exodermis of the rice roots (Extended Data Fig. S9)³⁵.”

The rice root spatial heatmap for *OPRIII* genes *Os06G0215500*, *Os06G0215600*, *Os06G0215900* and *Os06G0216300* (<http://spatialheatmap.baileyserreslab.org/>)³⁵ also revealed increased expression of the

OPR3 genes in the apical region, including the quiescent center (Extended Data Fig. S9). This information, combined with the time course of the wheat *OPR3* genes in the distal part of the seminal roots (Extended Data Fig. S2) and the developmentally regulated arrest of the RAM in 1RW and UBI::*OPR3* transgenic plants, suggest a role of the *OPR3* genes in the developmental regulation of the root stem cell niche in cereal plants.”

In addition, we performed a detailed subcellular localization of the wheat *OPR3* proteins *OPR3*-B1, *OPR3*-B2, *OPR3*-B3 and *OPR3*-A2 and demonstrated that they all have a cytoplasmic – nuclear localization. We added Extended Data Fig. S6 and the following section to the Results: “In *Arabidopsis*, *OPR3* (Group II) processes OPDA into OPC8 in the peroxisome, whereas *OPR2* (Group I) processes 4,5-ddh-JA into (-)-JA in the cytoplasm²⁷. Since wheat *OPR3* enzymes were able to use both OPDA and 4,5-ddh-JA as substrates *in vitro*, we performed a subcellular localization experiment in tobacco leaves using *OPR3*-GFP fusions and the mCherry peroxisome marker CD3-983²⁸. Our results indicate that *OPR3*-B1, *OPR3*-B2, *OPR3*-B3 and *OPR3*-A2 are all located in the nucleus and cytoplasm outside the peroxisome, similarly to what was previously reported for maize *OPR2* (Group III)²⁹ (Extended Data Fig. S6).”

Answers to Reviewer #2

Understanding the genetic bases of root architecture in bread wheat is thought to be essential to further improve crop production. Previously, Howell et al 2019 and Gabay et al 2021 have shown that the introgression of rye chromosome one (1RS) into bread wheat confers higher root biomass and a yield advantage under drought stress. Here, Gabay et al identify the underlying genes impacting the root length phenotype. A cluster of jasmonate (JA) biosynthesis genes called OPR3, impinge on root length, JA and ROS levels depending on their dosage. Low OPR3 copy number resulted in lower OPR3 dosage (or activity) and consequent longer roots/ reduced hormone and ROS levels, while higher OPR3 expression led to shorter roots, higher JA and ROS levels. Transcriptomics and network analyses were in line with these findings. Overall, the authors conclude that OPR3 can be used to modulate wheat root architecture. Findings are exciting and of broad general interest, the data are rigorous and the manuscript is well written and well presented.

MAJOR POINTS

1. The authors should perform drought assays to test if the modulation of root JA levels confers drought tolerance in wheat as expected from model plants. This can be done at the seedling stage in soil (eg. Kim et al 2017 Nat Plants) or hydroponic conditions with the use of eg. mannitol. Both positive or negative findings would greatly strengthen the impact of this work

Authors answer: We performed the suggested mannitol experiments and added the results in the new Figs. 1C and 1D and DataS1. Factorial ANOVAs including mannitol treatments and genotype showed significant effects for mannitol and genotype but no significant interactions between genotype and mannitol, both for root and leaf length and dry matter. In addition, the genotypes showed no significant differences in relative water content. Taken together, these results confirmed that the difference between 1RW and 1RS on drought tolerance in the field is most likely associated with their differences in root architecture and improved access to water at deeper soil layers, rather than to a differential physiological response to osmotic stress. We added: “To test if 1RS and 1RW exhibit differential physiological responses under osmotic stress, we compared them in hydroponic experiments with and without mannitol (200 mM). We found no significant differences between 1RS and 1RW for leaf length (Fig. 1C) and relative water content (Data S1), but detected significant reductions in root length (Fig. 1D) and root and shoot dry weight (Data S1) in 1RW relative to 1RS. All traits showed highly significant differences for the mannitol treatment, but no

significant Genotype x Mannitol interactions (Data S1), indicating that both genotypes responded similarly to the osmotic stress induced by mannitol (Fig. 1D).”

2. OPDA and JA are the precursors of bioactive JA-Ile, thus not active. Quantifications are provided only for JA (Fig.4). These should be complemented with OPDA and JA-Ile levels in mol/g FW and relative Limit of Quantification values for each compound.

Authors answer: Thank you for this suggestion. We performed the requested quantification of OPDA and JA-Ile and added the results in DataS14 and Figs. 4D and E. We detected significant increases of JA-Ile in both 1RW and UBI::OPRIII-R5 relative to 1RS, of similar magnitude to those presented before for JA. We detected no significant differences in OPDA, which may be related to the cytoplasmic/nuclear subcellular localization of the wheat OPRIII genes (new Extended Data Fig. S6). OPDA is the main substrate for the peroxisome located OPR genes, whereas 4,5-ddh-JA is the main substrate for the cytoplasmic OPRs. We added the following text to the results section: “We measured JA, JA-Ile and OPDA in the terminal 1 cm of the seminal roots at 6 DAG and observed a 2.5 to 3.3-fold increase in JA and JA-Ile concentrations in 1RW relative to 1RS ($P < 0.05$) and a 4.3 to 4.8-fold increase in UBI::OPRIII-R5 relative to 1RS ($P < 0.01$, Fig. 4D-E, Data S14). These results demonstrate that increases in *OPRIII* gene dosage or expression are associated with increases in JA and JA-Ile in early seminal root development. We did not find significant differences in OPDA among the three genotypes (Data S14).”.

MINOR POINTS

- Fig.5C. Is there a specific reason for root tips facing upwards? If not, I would recommend to orient the meristems downwards for consistency with all other figures in the paper.

Authors answer: modified as requested by the reviewer

Answers to Reviewer #3

This paper is an excellent contribution to our understanding of root growth regulation, supported by strong evidence from a series of experiments that cover multiple ways of demonstrating the identity and function of the identified genes that are responsible for the observed phenotypes. Furthermore, there are important practical implications of the findings for breeding improved wheat varieties for water-limited environments. I would recommend publication.

This paper extends the research reported by the lead author and others in a 2021 paper (cited), which reported the discovery that gene dosage effects explained the reduction in root growth observed in wheat lines containing a recombinant 1BS.1BL chromosomal translocation from rye that contains an interstitial wheat introgression in the distal region of the translocated rye chromosome (1RW). This report is a continuation of an elegant and painstakingly careful pathway of discovery into the genetic basis of this diminished root growth phenotype, eliminating the most obvious hypothesised causes (loss of 1RS rye genes or added wheat 1BS genes), and concluding that the 1RW introgression was actually quite complex structurally, with multiple gene duplications, causing the observed gene dosage effect. The previous paper identified 14 high confidence candidate genes in the chromosomal region that remained after radiation-induced deletion of two of the duplicated rye segments and two wheat segments in the distal region, leaving 2 duplicated rye segments responsible for the dosage effect. From that list, OPRIII was highlighted as a likely candidate. The current paper identifies the duplicated gene within

that region responsible for the root growth inhibition of 1RW compared with the original IRS, which through a combination of approaches was confirmed to be the OPRIII gene family.

In addition, experiments described a functional characterisation of expressed OPRIII genes via NADPH consumption assays using OPDA (jasmonate precursor) as substrate. The results were consistent with oxophytodienoate reductase activity. A series of experiments were conducted to confirm the role of JA in modulating the reduced root growth of lines with increased OPRIII gene number by treating plants with ibuprofen (IBU), which inhibits jasmonyl-isoleucine synthesis, the active form of the phytohormone. Results supported the role of JA by nearly recovering the long root phenotype in the IBU-treated 1RW line, which contains duplicated OPRIII genes. Levels of JA were also elevated in root tips of 1RW compared with IRS (two OPRIII genes from rye), and were further elevated in the 1RW lines that overexpressed the rye OPRIII (R5). All experiments were repeated and there was appropriate and thorough statistical analysis throughout.

Minor comments:

1. P4L15 and L38: Authors point out that ROS play important roles in the meristem, although the effect on root growth was also likely due to altered cell expansion in the elongation zone. There is evidence that ROS, and in particular glutathione, are important components of the root growth regulatory pathway in maize (Kang et al. 2022 <https://doi.org/10.3390/antiox11050820>). Unless authors are certain that the site of action is exclusively in the meristem, mention of this additional role of ROS would help make the story more complete.

Author answers: Thank you we added the comment and the Kang et al. (2022) reference to the Discussion section: “There is also evidence that ROS, and in particular glutathione, are important components of the root growth regulatory pathway in maize affecting both the meristematic and elongation zones ³⁴.”

2. Authors showed in the 2021 paper that the IRS duplication had a smaller effect on root growth than the OPRIII-B1 duplication. It would be helpful if authors explained why therefore the OPRIII-B1 gene was chosen for editing, rather than OPRIII-R5, which was not edited, but used in the overexpression lines (along with B1 as well).

Authors answer: In our initial CRISPR experiment, we only found mutations in the OPRIII-B1 genes in spite of targeting both wheat and rye OPRIII genes. We generated an additional set of CRISPR transgenic lines in which we were able to identify two deletions generating premature stop codons on gene OPRIII-R1. We included the new results in the revised version, which are consistent with those initially reported for OPRIII-B1.

We added new Fig. 2B and the following text to the Results section: “To validate the role of the OPRIII genes on seminal root length, we targeted the duplicated wheat and rye OPRIII genes in 1RW for CRISPR-Cas9 editing. We found a 32-bp frame-shift induced deletion in OPRIII-B1 exon 2 (mut-OPRIII-B1) and two deletions in OPRIII-R1 of 7-bp (mut-OPRIII-R1.7) and 3-bp (mut-OPRIII-R1.3).”

and

“The T2 plants carrying at least one copy of the mut-OPRIII-R1.7 and mut-OPRIII-R1.3 deletions also showed significantly longer roots than the 1RW sister lines (Fig. 2B, 17-23% increase).”

3. *Use of the ubiquitin promoter, which achieved constitutive over-expression of OPRIII genes including in leaves (P3,L25), could have resulted in poor shoot growth, which could have affected root growth towards the end of the experiment as roots shifted from drawing on seed reserves to receiving photoassimilate from the shoot. Hence the gene effect in O/E lines could have been due to a shoot effect rather than or in addition to a direct effect on roots. This seems an unlikely explanation of the short root phenotype, but perhaps some note on general appearance of the shoots would be warranted.*

Authors answer: We added two experiments describing the effect of the increased OPRIII dosage in 1RW relative to 1RS on the aerial part (Table 1): “A combined ANOVA for two experiments showed no significant differences in leaf number, length of the youngest fully expanded leaf or aerial dry weight at 26 DAG (Table 1).”

We then characterized the effect of the transgene on the aerial part and observed a significant effect of the transgene on both the root and aerial dry weight, although the ratio between the two was not significantly affected. We agree with the reviewer that we need to include a disclaimer that we cannot rule out a contribution of the aerial part to the observed differences in the roots in the transgenic plants overexpressing the OPRIII genes. However, we also clarified that when the OPRIII genes are under the control of their native promoters, they are expressed mainly in the roots, suggesting that their primary effect is in this tissue.

We added the new results in Data S9 and S10 and a new sentence to the Results section describing the effect of the transgene: “Since ectopic expression of these two OPRIII genes resulted in significant reductions in leaf length (9-10%) and aerial dry weight (30-36%) at 23 DAG (Data S9-10), we cannot rule out an indirect effect of these changes on the observed differences in the roots of the transgenic plants. However, when driven by their natural promoters, the OPRIII genes are expressed mainly in the roots, suggesting a primary effect in this tissue.”

4. *In the Introduction it was mentioned that introgression of the 1BS genes into the 1RS rye translocation was done to improve grain quality (e.g. dough strength), but this also resulted in reduced root growth, or at least loss of the improved root growth seen in the intact 1RS. There should be a comment on how it might be possible to preserve the improved quality of the recombinant 1RS translocation while eliminating the duplicated OPRIII genes responsible for deleterious root growth. In the paper it is not made clear if the parental line used in this study was one described by Lukaszewski (2000) that included the wheat glutenin/gliadin genes rather than the line that removed the rye secalin gene to improve quality. Either way, knowledge of the identity and location of the OPRIII duplications should allow targeted selections to maintain root growth and grain quality in parallel. A comment would be warranted.*

Authors answer: This is a good suggestion. Indeed, once CRISPR wheat lines become less controversial, OPRIII mutants generated by CRISPR in 1RW can be used to restore root growth in a background with improved breadmaking quality. The engineered distal duplicated segment will then need to be combined again with the proximal wheat segment to restore full breadmaking functionality. We added to the discussion: “In addition to the basic biological questions that can be investigated by the manipulation of the OPRIII genes, this study also points to potential practical applications. One example is the increased root length in the 1RW CRISPR-Cas9 mutants, which provides a path to restore the good performance of the 1RS lines under water stress to the 1WW line (Fig. 1A) with improved breadmaking quality 10, 11. However, the seminal roots of the single gene mutants were still shorter than the original 1RS line (Fig. 2B), suggesting that the OPRIII gene dosage may need to be fine-tuned

to restore root growth and grain yield potential to the 1RS levels”

We also clarified in the introduction which 1BS segment was replacing the rye secalins that confer sticky dough, and which one was restoring the Gli-B1/Glu-B3 locus associated with good breadmaking quality: “We introgressed the engineered 1WW chromosome into the 1RS.1BL cv. Hahn by six backcrosses, and then generated three lines with either the complete 1RS (1RS), the proximal 1BS introgression removing the secalins (1WR), or the distal 1BS introgression restoring the Gli-B1 / Glu-B3 locus (1RW, Fig. 1A).”

5. P3,L36 Authors should consider wording that takes into account that NBT staining is not specific for superoxide, as NBT apparently can be reduced by other enzymatic reductases present in the tissue. However, it unlikely that a different result would be obtained or that conclusions would be altered by using a different method of quantification.

Authors answer: We have revised wording and substituted anion superoxides by ROS throughout the paper and Online Methods.

6. Fig. 5: the ordering of Figures, or the lettering order for B and C needs checking.

Authors answer: This error was corrected.

7. A possible point of confusion for readers is why roots of 1RS lines grow faster than roots of plants without the rye translocation, with the original 1BS chromosome. Is this because the intact 1RS chromosome contains only two OPRIII genes (R1 and R5; Fig. 1E), whereas the 1BS chromosome contains four (B1, B2, B3, B9), hence greater dosage, even though the 1RS genes had a stronger effect on root growth than the 1BS genes (Gabay et al 2021)?

Authors answer: We do not have Hahn isogenic lines with the complete 1BS chromosome to make this comparison. However, in Gabay et al. 2021, we compared the 1RS line with a sister line T-18 carrying a longer distal 1BS translocations but without the 1RS duplication. The root growth rates during the experiment were similar between T-18 and 1RS. We clarified this in the expanded introduction: “The previous study¹⁷, also showed that different 1RS-1BS recombinant lines with longer distal regions from 1BS or 1RS (but without the duplication present in 1RW) have long seminal roots, suggesting that changes in gene dosage are responsible for the shorter seminal roots in 1RW rather than specific 1BS or 1RS genes.”

8. The results show early effects of JA accumulation (via OPRIII activity) on seminal root growth that decay over short periods of time. Sustained root growth in seminal and crown roots must occur in the field and soil-based pot experiments that have shown greater root biomass or root depth of 1RS lines compared with lines without the rye translocation. It is not clear how to connect these seedling results to those observed on more mature root systems.

Authors answer: We added measurements of the nodal roots, which showed a similar developmentally regulated growth arrest in 1RW relative to 1RS as the seminal roots. The new experiments also tended to

show a higher root dry weight biomass in 1RS than in 1RW. These results are consistent with the reduced root density observed in 1RW at the field below 20 cm in our previous study. We added a panel to Fig. 1 comparing the growth of the nodal roots in 1RS and 1RW and the following sentence to the results section: “Nodal roots started to appear at 18 DAG and initially grew similarly in 1RS and 1RW. However, by 27 DAG the 1RS longest nodal roots were 2.7 cm longer ($P = 0.031$) than the equivalent root in 1RW, a difference that increased to 10.8 cm at 31 DAG and to 18.1 cm at 34 DAG ($P < 0.001$, Fig. 1B). These results indicate that the 1RW nodal roots experience a similar developmentally-regulated growth arrest as the seminal roots. Taken together, these results are consistent with the higher root density and increased water access of the 1RS plants relative to 1RW in the field experiments¹⁴.”

9. The authors state at the conclusion of the paper that manipulating OPRIII gene dosage could be a way to engineer improved root systems. However, this study focused on reducing the effects of extraneous dosage responsible for reduced root growth, and targeted deletions (or IBU treatment) were effective only in returning root growth back to wildtype (1RS) rates. Could authors speculate on how further increases in root growth rate (not reductions) would be possible through breeding, either in a 1RS background, or in lines without the rye translocation and only containing wheat OPRIII genes? It seems likely that total elimination of root-specific JA production would go too far, resulting in aberrant control of ROS homeostasis in the root growth zone (and perhaps loss of beneficial secondary effects of JA-mediated protection of roots against root herbivory and soil pathogens).

Authors answer: The CRISPR-Cas9 mutants showed that changes in a single OPRIII gene can result in seminal roots longer than in 1RW. We added in the discussion: “...However, the seminal roots of the single gene mutants were still shorter than the original 1RS line (Fig. 2B), suggesting that the OPRIII gene dosage may need to be fine-tuned to restore root growth and grain yield potential to the 1RS levels”.

We then noted in the Discussion that very divergent haplotypes including different number of functional OPRIII genes were found within the limited sample of sequenced wheat genomes and exome captures. The next step in our research will be the characterization of the extensive natural variation discovered in the OPRIII loci and the evaluation of their effects on root architecture. We are not only interested in the discovery of longer roots, but on the variations in root architecture that can help wheat adapt to different types of soils. In addition to the characterization of the natural variation we will continue the characterization of different OPRIII induced mutants to explore their potential role on fine-tuning root growth rates and architecture. We added at the end of the Discussion: “...the extensive variation detected in the number of functional OPRIII genes in the available sequenced wheat genomes (Data S2) suggests that natural variation in these genes may have contributed to the adaptation of wheat to different soil environments. The identification of OPRIII gene dosage as a sensitive point in the regulation of the JA-biosynthetic pathway provides a previously unknown target to engineer root architecture in wheat and possibly other cereal crops”.

Reviewers' Comments:

Reviewer #1:

Remarks to the Author:

Understanding the genetic regulation of root traits in crops like wheat is vital for ongoing food security efforts. The current paper (Gabay et al.) characterises 1RW wheat plants encoding an engineered chromosome that exhibit slow growing roots as compared to the original 1RS rye introgression line. The authors have elegantly demonstrated that the short root phenotypes of 1RW line is due to a higher gene dosage of OPR JA synthesis genes by showing (1) mutations in OPRs lead to longer root phenotypes and (2) over-expression of OPR leads to shorter root phenotypes. The authors have also measured JA content and established high JA accumulation leads to shorter root phenotypes, whilst pharmacological treatments blocking JA synthesis rescues root length.

The revised manuscript's conclusions are backed by up by the experimental evidence presented and, in response to reviewers feedback, changes to the original text. The authors have done a good job addressing all 3 reviewers comments and deserve credit for the quality and significance of the results reported in the revised manuscript.

Reviewer #2:

Remarks to the Author:

The authors have addressed my main concerns. In addition, they have replied convincingly to the other two reviewers by generating new data and improving the text. The phenotypic effects of the monocot-specific OPRIII family on wheat root architecture extends far beyond what is known in Arabidopsis. These are breakthrough findings for both fundamental and applied research

The authors should report OPDA/JA/JA-Ile levels (Fig 4&S14) in molar units per g fresh weight (eg. pmol/g FW), and not in peak area given that standards were used. In methods L229, deuterated standards used should be specified, and Limit of quantification (LOQ) reported for each of the three molecules at the end of that methods section.

A few minor points the authors may consider:

- L51. Reference missing?
- L200. Driven
- L225. A general explanation for OPDA/JA/JA-Ile would be helpful here. Eg. 'We measured levels of bioactive JA-Ile and its precursors JA and OPDA in the terminal..', or similar.
- L305. 12-OXOPHYTODIENOATE REDUCTASE should not be in italics as it refers to an enzyme

Reviewer #3:

Remarks to the Author:

The authors have responded to each of the points in the review and have either provided a satisfactory response, or have made changes that have resolved the issue raised. The additional experiments have further strengthened the paper. I have no further comments or questions for the revised manuscript; it looks fine and is an excellent contribution. It appears from the authors' responses that points raised by the other reviewers have been dealt with satisfactorily.

Reviewer #1

Understanding the genetic regulation of root traits in crops like wheat is vital for ongoing food security efforts. The current paper (Gabay et al.) characterises IRW wheat plants encoding an engineered chromosome that exhibit slow growing roots as compared to the original IRS rye introgression line. The authors have elegantly demonstrated that the short root phenotypes of IRW line is due to a higher gene dosage of OPR JA synthesis genes by showing (1) mutations in OPRs lead to longer root phenotypes and (2) over-expression of OPR leads to shorter root phenotypes. The authors have also measured JA content and established high JA accumulation leads to shorter root phenotypes, whilst pharmacological treatments blocking JA synthesis rescues root length. The revised manuscript's conclusions are backed by up by the experimental evidence presented and, in response to reviewers feedback, changes to the original text. The authors have done a good job addressing all 3 reviewers comments and deserve credit for the quality and significance of the results reported in the revised manuscript.

Authors answer: We appreciate all the reviewer's comments and we are thankful for their contribution to improve this manuscript.

Reviewer #2

The authors have addressed my main concerns. In addition, they have replied convincingly to the other two reviewers by generating new data and improving the text. The phenotypic effects of the monocot-specific OPRIII family on wheat root architecture extends far beyond what is known in Arabidopsis. These are breakthrough findings for both fundamental and applied research. The authors should report OPDA/JA/JA-Ile levels (Fig 4&S14) in molar units per g fresh weight (eg. pmol/g FW), and not in peak area given that standards were used. In methods L229, deuterated standards used should be specified, and Limit of quantification (LOQ) reported for each of the three molecules at the end of that methods section.

Authors answer: Because deuterated/labeled OPDA and JA-Ile standards are not available at the UC Riverside Metabolomics facility, we have no other alternative but to base our measurements on the peak area values. These standards are not easy to obtain from commercial sources and we are unable to synthesize them. We believe that the provided peak area values are as reliable as the pmol/g FW for relative comparative purposes among lines.

A few minor points the authors may consider:

- L51. Reference missing?

Authors answer: We added reference #4 at the end of the requested sentence: “This has prompted new efforts to understand and modify wheat root architecture to optimize water acquisition in both common (*Triticum aestivum*, genomes AABBDD) and durum wheat (*T. turgidum* ssp. *durum*, genomes AABB) ⁴”.

- L200. Driven.

Authors answer: Modified as requested: “However, when driven by their natural promoters, the OPRIII genes are expressed mainly in the roots, suggesting a primary effect in this tissue.”

- L225. A general explanation for OPDA/JA/JA-Ile would be helpful here. Eg. ‘We measured levels of bioactive JA-Ile and its precursors JA and OPDA in the terminal..’, or similar.

Authors answer: Modified as requested: “We measured levels of the bioactive JA-Ile and its precursors JA and OPDA in the terminal 1 cm of the seminal roots at 6 DAG.”

- L305. 12-OXOPHYTODIENOATE REDUCTASE should not be in italics as it refers to an enzyme

Authors answer: Modified to non-italics: “We show here that the little studied genes from the monocot-specific *OPR3* subfamily encode cytoplasmic and nuclear 12-OXOPHYTODIENOATE REDUCTASE enzymes that regulate a critical step in the synthesis of JA-Ile”

Reviewer #3 (Remarks to the Author):

The authors have responded to each of the points in the review and have either provided a satisfactory response, or have made changes that have resolved the issue raised. The additional experiments have further strengthened the paper. I have no further comments or questions for the revised manuscript; it looks fine and is an excellent contribution. It appears from the authors' responses that points raised by the other reviewers have been dealt with satisfactorily.

Authors answer: We appreciate all the reviewer's comments and we are thankful for their contribution to improve this manuscript.